# Logarithmic Regret in Feature-based Dynamic Pricing

**Jianyu Xu**
Department of Computer Science
University of California, Santa Barbara
Santa Barbara, CA 93106
xu_jy15@ucsb.edu

**Yu-Xiang Wang**
Department of Computer Science
University of California, Santa Barbara
Santa Barbara, CA 93106
yuxiangw@cs.ucsb.edu

## Abstract

Feature-based dynamic pricing is an increasingly popular model of setting prices for highly differentiated products with applications in digital marketing, online sales, real estate and so on. The problem was formally studied as an online learning problem [Javanmard and Nazerzadeh, 2019] where a seller needs to propose prices *on the fly* for a sequence of $T$ products based on their features $x$ while having a small *regret* relative to the best —"omniscient"— pricing strategy she could have come up with in hindsight. We revisit this problem and provide two algorithms (EMLP and ONSP) for stochastic and adversarial feature settings, respectively, and prove the optimal $O(d \log T)$ regret bounds for both. In comparison, the best existing results are $O\left(\min\left\{\frac{1}{\lambda_{\min}^2} \log T, \sqrt{T}\right\}\right)$ and $O(T^{2/3})$ respectively, with $\lambda_{\min}$ being the smallest eigenvalue of $\mathbb{E}[xx^T]$ that could be arbitrarily close to 0. We also prove an $\Omega(\sqrt{T})$ information-theoretic lower bound for a slightly more general setting, which demonstrates that "knowing-the-demand-curve" leads to an exponential improvement in feature-based dynamic pricing.

## 1 Introduction

The problem of pricing — to find a high-and-acceptable price — has been studied since Cournot [1897]. In order to locate the optimal price that maximizes the revenue, a firm may adjust their prices of products frequently, which inspires the *dynamic pricing* problem. Existing works [Kleinberg and Leighton, 2003, Broder and Rusmevichientong, 2012, Chen and Farias, 2013, Besbes and Zeevi, 2015] primarily focus on pricing a single product, which usually will not work well in another setting when thousands of new products are being listed every day with no prior experience in selling them. Therefore, we seek methods that approach an acceptable-and-profitable price with only observations on this single product and some historical selling records of other products.

In this work, we consider a "feature-based dynamic pricing" problem, which was studied by Amin et al. [2014], Cohen et al. [2020], Javanmard and Nazerzadeh [2019]. In this problem setting, a sales session (product, customer and other environmental variables) is described by a feature vector, and the customer's *expected* valuation is modeled as a linear function of this feature vector.

---

Feature-based dynamic pricing. For $t = 1, 2, ..., T$ :
1. A feature vector $x_t \in \mathbb{R}^d$ is revealed that describes a sales session (product, customer and context).
2. The customer valuates the product as $w_t = x_t^\top \theta^* + N_t$.
3. The seller proposes a price $v_t > 0$ concurrently (according to $x_t$ and historical sales records).
4. The transaction is successful if $v_t \le w_t$, i.e., the seller gets a reward (payment) of $r_t = v_t \cdot \mathbb{1}(v_t \le w_t)$.

---

Here $T$ is unknown to the seller (and thus can go to infinity), $x_t$'s can be either stochastic (e.g., each sales session is drawn i.i.d.) or adversarial (e.g., the sessions arrive in a strategic sequence), $\theta^* \in \mathbb{R}^d$ is a fixed parameter for all time periods, $N_t$ is a zero-mean noise, and $\mathbb{1}_t = \mathbb{1}(v_t \le w_t)$ is an indicator that equals 1 if $v_t \le w_t$ and 0 otherwise. In this online-fashioned setting, we only see

Table 1: Related Works and Regret Bounds w.r.t. $T$

| Algorithm | Work | Regret (upper) bound | Feature | Noise |
|---|---|---|---|---|
| LEAP | [Amin et al., 2014] | $\tilde{O}(T^{\frac{2}{3}})$ | i.i.d. | Noise-free |
| EllipsoidPricing | [Cohen et al., 2020] | $O(\log T)$ | adversarial | Noise-free |
| EllipsoidEXP4 | [Cohen et al., 2020] | $\tilde{O}(T^{\frac{2}{3}})$ | adversarial | Sub-Gaussian |
| PricingSearch | [Leme and Schneider, 2018] | $O(\log \log(T))$ | adversarial | Noise-free |
| RMLP | [Javanmard and Nazerzadeh, 2019] | $O(\log T / C_{\min}^2)^\dagger$ $O(\sqrt{T})$ | i.i.d. | Log-concave, distribution-known |
| RMLP-2 | [Javanmard and Nazerzadeh, 2019] | $O(\sqrt{T})$ | i.i.d. | Known parametric family of log-concave. |
| ShallowPricing | [Cohen et al., 2020] | $O(poly(\log T))$ | adversarial | Sub-Gaussian, known $\sigma = O(\frac{1}{T \log T})$ |
| CorPV | [Krishnamurthy et al., 2021] | | | |
| Algorithm 2 (MSPP) | [Liu et al., 2021] | $O(\log \log(T))$ | adversarial | Noise-free |
| **EMLP** | This paper | $O(\log T)$ | i.i.d. | Strictly log-concave, distribution-known |
| **ONSP** | This paper | $O(\log T)$ | adversarial | Strictly log-concave, distribution-known |

$^\dagger$ $C_{\min}$ is the restricted eigenvalue condition. It reduces to the smallest eigenvalue of $\mathbb{E}[xx^\top]$ in the low-dimensional case we consider.

and sell one product at each time. Also, the feedback is *Boolean Censored*, which means we can only observe $\mathbb{1}_t$ instead of knowing $w_t$ directly. The best pricing policy for this problem is the one that maximizes the *expected* reward, and the *regret* of a pricing policy is accordingly defined as the difference of expected rewards between this selected policy and the best policy.

**Summary of Results.** Our contributions are threefold.

1. When $x_t$'s are independently and identically distributed (i.i.d.) from an unknown distribution, we propose an "Epoch-based Max-Likelihood Pricing (EMLP)" algorithm that guarantees a regret bound at $O(d \log T)$. The design of EMLP is similar to that of the RMLP algorithm in Javanmard and Nazerzadeh [2019], but our new analysis improves their regret bound at $O(\sqrt{T})$ when $\mathbb{E}[xx^\top]$ is near singular.

2. When $x_t$'s are adversarial, we propose an "Online-Newton-Step Pricing (ONSP)" algorithm that achieves $O(d \log T)$ regret on constant-level noises for the first time, which exponentially improves the best existing result of $O(T^{2/3})$ [Cohen et al., 2020].[1]

3. Our methods that achieve logarithmic regret require knowing the exact distribution of $N_t$ in advance, as is also assumed in Javanmard and Nazerzadeh [2019]. We prove an $\Omega(\sqrt{T})$ lower bound on the regret if $N_t \sim \mathcal{N}(0, \sigma^2)$ where $\sigma$ is *unknown*, even with $\theta^*$ given and $x_t$ fixed for all $t$.

The $O(\log T)$ regret of EMLP and ONSP meets the information-theoretical lower bound [Theorem 5, Javanmard and Nazerzadeh, 2019]. In fact, the bound is optimal even when $w_t$ is revealed to the learner [Mourtada, 2019]. From the perspective of characterizing the hardness of dynamic pricing problems, we generalize the classical results on "The Value of Knowing a Demand Curve" [Kleinberg and Leighton, 2003] by further dividing the random-valuation class with an exponential separation of: (1) $O(\log T)$ regret for knowing the *demand curve* exactly (even with adversarial features), and (2) $\Omega(\sqrt{T})$ regret for *almost* knowing the *demand curves* (up to a one-parameter parametric family).

## 2 Related Works

In this section, we discuss our results relative to existing works on feature-based dynamic pricing, and highlight the connections and differences to the related settings of contextual bandits and contextual search (for a broader discussion, see Appendix A).

**Feature-based Dynamic Pricing.** There is a growing body of work on dynamic pricing with linear features [Amin et al., 2014, Qiang and Bayati, 2016, Cohen et al., 2020, Javanmard and Nazerzadeh, 2019]. Table 1 summarizes the differences in the settings and results[2]. Among these work, our paper directly builds upon [Cohen et al., 2020] and [Javanmard and Nazerzadeh, 2019], as we share the same setting of online feature vectors, linear and noisy valuations and Boolean-censored feedback. Relative to the results in [Javanmard and Nazerzadeh, 2019], we obtain $O(d \log T)$ regret under weaker assumptions on the sequence of input features — in both distribution-free stochastic feature

---

[1]Previous works [Cohen et al., 2020, Krishnamurthy et al., 2021] did achieve polylog regrets, but only for negligible noise with $\sigma = O(\frac{1}{T \log T})$.

[2]We only concern the dependence on $T$ since there are various different assumptions on $d$.

setting and the adversarial feature setting. It is to be noted that [Javanmard and Nazerzadeh, 2019] also covers the sparse high-dimensional setting, and handles a slightly broader class of demand curves. Relative to [Cohen et al., 2020], in which the adversarial feature-based dynamic pricing was first studied, our algorithm ONSP enjoys the optimal $O(d \log T)$ regret when the noise-level is a constant. In comparison, Cohen et al. [2020] reduces the problem to contextual bandits and applies the (computationally inefficient) "EXP-4" algorithm [Auer et al., 2002] to achieve a $\tilde{O}(T^{2/3})$ regret. The "bisection" style-algorithm in both Cohen et al. [2020] and Krishnamurthy et al. [2021] could achieve $\tilde{O}(poly(d)poly \log(T))$ regrets but requires a small-variance subgaussian noise satisfying $\sigma = O(\frac{1}{T \log T})$.

**Lower Bounds.** Most existing works focus on the lower regret bounds of non-feature-based models. Kleinberg and Leighton [2003] divides the problem setting as fixed, random, and adversarial valuations, and then proves each a $\Theta(\log \log T)$, $\Theta(\sqrt{T})$, and $\Theta(T^{2/3})$ regret, respectively. Broder and Rusmevichientong [2012] further proves a $\Theta(\sqrt{T})$ regret in general parametric valuation models. In this work, we generalize the methods of Broder and Rusmevichientong [2012] to our feature-based setting and further narrow it down to a linear-feature Gaussian-noisy model. As a complement to Kleinberg and Leighton [2003], we further separate the exponential regret gap between: (1) $O(\log T)$ of the hardest (adversarial feature) totally-parametric model, and (2) $\Omega(\sqrt{T})$ of the simplest (fixed known expectation) unknown-$\sigma$ Gaussian model.

**Contextual Bandits.** For readers familiar with the online learning literature, our problem can be reduced to a contextual bandits problem [Langford and Zhang, 2007, Agarwal et al., 2014] by discretizing the prices. But this reduction only results in $O(T^{2/3})$ regret, as it does not capture the special structure of the feedback: *an accepted price indicates the acceptance of all lower prices*, and vise versa. Moreover, when comparing to linear bandits [Chu et al., 2011], it is the valuation instead of the expected reward that we assume to be linear.

**Contextual Search.** Feature-based dynamic pricing is also related to the contextual search problem [Lobel et al., 2018, Leme and Schneider, 2018, Liu et al., 2021, Krishnamurthy et al., 2021], which often involves learning from Boolean feedbacks, sometimes with a "pricing loss" and "noisy" feedback. These shared jargons make this problem *appearing* very similar to our problem. However, except for the noiseless cases [Lobel et al., 2018, Leme and Schneider, 2018], contextual search algorithms, even with "pricing losses" and "Noisy Boolean feedback" [e.g., Liu et al., 2021], do *not* imply meaningful regret bounds in our problem setup due to several subtle but important differences in the problem settings. Specifically, the noisy-boolean feedback model of [Liu et al., 2021] is about randomly toggling the "purchase decision" determined by the *noiseless* valuation $x^\top \theta^*$ with probability $0.5 - \epsilon$. This is incompatible to our problem setting where the purchasing decision is determined by a noisy valuation $x^\top \theta^* +$ Noise. Ultimately, in the setting of [Liu et al., 2021], the optimal policy alway plays $x^\top \theta^*$, but our problem is harder in that we need to exploit the noise and the optimal price could be very different from $x^\top \theta^*$. [3] Krishnamurthy et al. [2021] also discussed this issue explicitly and considered the more natural noisy Boolean feedback model studied in this paper. Their result, similar to Cohen et al. [2020], only achieves a logarithmic regret when the noise on the valuation is vanishing in an $\tilde{O}(1/T)$ rate.

## 3 Problem Setup

**Symbols and Notations.** Now we introduce the mathematical symbols and notations involved in the following pages. The game consists of $T$ rounds. $x_t \in \mathbb{R}^d$, $v_t \in \mathbb{R}_+$ and $N_t \in \mathbb{R}$ denote the feature vector, the proposed price and the noise respectively at round $t = 1, 2, ..., T$. [4] We denote the product $u_t := x_t^\top \theta^*$ as an *expected valuation*. At each round, we receive a payoff (reward) $r_t = v_t \cdot \mathbb{1}_t$, where the binary variable $\mathbb{1}_t$ indicates whether the price is accepted or not, i.e., $\mathbb{1}_t = \mathbf{1}(v_t \leq w_t)$. As we may estimate $\theta^*$ in our algorithms, we denote $\hat{\theta}_t$ as an estimator of $\theta^*$, which we will formally define in the algorithms. Furthermore, we denote some functions that are related to noise distribution: $F(\omega)$ and $f(\omega)$ denote the cumulative distribution function (CDF) and probability density function (PDF) sequentially. We know that $F'(\omega) = f(\omega)$ if we assume differentiability. To concisely denote all data observed up to round $\tau$ (i.e., feature, price and payoff of all past rounds), we define

---

[3]As an explicit example, suppose the valuation $x^\top \theta^* = 0$, then the optimal price must be $> 0$ in order to avoid zero return.

[4]In an epoch-design situation, a subscript $(k, t)$ indicates round $t$ of epoch $k$.

$hist(\tau) = \{(x_t, v_t, \mathbb{1}_t) \text{ for } t = 1, 2, ..., \tau\}$. $hist(\tau)$ represents the *transcript* of all observed random variables before round $(\tau + 1)$.

We define

$$l_t(\theta) := -\mathbb{1}_t \cdot \log\left(1 - F(v_t - x_t^\top \theta)\right) - (1 - \mathbb{1}_t) \log\left(F(v_t - x_t^\top \theta)\right) \tag{1}$$

as a negative log-likelihood function at round $t$. Also, we define an expected log-likelihood function $L_t(\theta)$:

$$L_t(\theta) := \mathbb{E}_{N_t}[l_t(\theta)|x_t] \tag{2}$$

Notice that we will later define an $\hat{L}_k(\theta)$ which is, however, not an expectation.

**Definitions of Key Quantities.** We firstly define an *expected reward* function $g(v, u)$.

$$g(v, u) := \mathbb{E}[r_t|v_t = v, x_t^\top \theta^* = u] = v \cdot P[v \le x_t^\top \theta^* + N_t] = v \cdot (1 - F(v - u)). \tag{3}$$

This indicates that if the expected valuation is $u$ and the proposed price is $v$, then the (conditionally) expected reward is $g(v, u)$. Now we formally define the *regret* of a policy (algorithm) $\mathcal{A}$ as is promised in Section 1.

**Definition 1** (Regret). *Let* $\mathcal{A} : \mathbb{R}^d \times \left(\mathbb{R}^d, \mathbb{R}, \{0, 1\}\right)^{t-1} \to \mathbb{R}$ *be a policy of pricing, i.e.* $\mathcal{A}(x_t, hist(t - 1)) = v_t$. *The regret of* $\mathcal{A}$ *is defined as follows.*

$$Reg_{\mathcal{A}} = \sum_{t=1}^{T} \max_v g(v, x_t^\top \theta^*) - g(\mathcal{A}(x_t, hist(t - 1)), x_t^\top \theta^*). \tag{4}$$

*Here $hist(t - 1)$ is the historical records until $(t - 1)^{th}$ round.*

**Summary of Assumptions.** We specify the problem settings by proposing three assumptions.

**Assumption 1** (Known, bounded, strictly log-concave distribution). *The noise $N_t$ is independently and identically sampled from a distribution whose CDF is $F$. Assume that $F \in \mathbb{C}^2$ is strictly increasing and that $F$ and $(1 - F)$ are strictly log-concave. Also assume that $f$ and $f'$ are bounded, and denote $B_f := \sup_{\omega \in \mathbb{R}} f(\omega)$, $B_{f'} := \sup_{\omega \in \mathbb{R}} |f'(\omega)|$ as two constants.*

**Assumption 2** (Bounded convex parameter space). *The true parameter $\theta^* \in \mathbb{H}$, where $\mathbb{H} \subseteq \{\theta : ||\theta||_2 \le B_1\}$ is a bounded convex set and $B_1$ is a constant. Assume $\mathbb{H}$ is known to us (but $\theta^*$ is not).*

**Assumption 3** (Bounded feature space). *Assume $x_t \in D \subseteq \{x : ||x||_2 \le B_2\}, \forall t = 1, 2, \ldots, T$. Also, $0 \le x^\top \theta \le B, \forall x \in D, \forall \theta \in \mathbb{H}$, where $B = B_1 \cdot B_2$ is a constant.*

Assumption 2 and 3 are mild as we can choose $B_1$ and $B_2$ large enough. In Section 4.1, we may add further complement to Assumption 3 to form a stochastic setting. Assumption 1 is stronger since we might not know the exact CDF in practice, but it is still acceptable from an information-theoretic perspective. There are at least three reasons that lead to this assumption: Primarily, this is necessary if we hope to achieve an $O(\log(T))$ regret. We will prove in Section 5.3 that an $\Omega(\sqrt{T})$ is unavoidable if we cannot know one parameter exactly. Moreover, the pioneering work of Javanmard and Nazerzadeh [2019] also assumes a known noise distribution with log-concave CDF, and many common distributions are actually strictly log-concave, such as Gaussian and logistic.[5] Besides, although we did not present a method to precisely estimate $\sigma$ in this work, it is a reasonable algorithm to replace with a plug-in estimator estimated using historical offline data. As we have shown, not knowing $\sigma$ requires $O(\sqrt{T})$ regret in general, but the lower bound does not rule out the plug-in approach achieving a smaller regret for interesting subclasses of problems in practice.

Finally, we state a lemma and define an argmax function helpful for our algorithm design.

**Lemma 2** (Uniqueness). *For any $u \ge 0$, there exists a unique $v^* \ge 0$ such that $g(v^*, u) = \max_{v \in \mathbb{R}} g(v, u)$. Thus, we can define a* greedily pricing *function that maximizes the expected reward:*

$$J(u) := \arg\max_v g(v, u) \tag{5}$$

Please see the proof of Lemma 2 in Appendix B.1.

## 4 Algorithms

In this section, we propose two dynamic pricing algorithms: EMLP and ONSP, for stochastic and adversarial features respectively.

---

[5]In fact, $F$ and $(1 - F)$ are both log-concave if its PDF is log-concave, according to Prekopa's Inequality.

**Algorithm 1** Epoch-based max-likelihood pricing (EMLP)

> **Input:** Convex and bounded set $\mathbb{H}$
> Observe $x_1$, randomly choose $v_1$ and get $r_1$.
> Solve $\hat{\theta}_1 = \arg\min_{\theta \in \mathbb{H}} l_1(\theta)$;
> **for** $k = 1$ **to** $\lfloor \log_2 T \rfloor + 1$ **do**
> > Set $\tau_k = 2^{k-1}$;
> > **for** $t = 1$ **to** $\tau_k$ **do**
> > > Observe $x_{k,t}$;
> > > Set price $v_{k,t} = J(x_{k,t}^\top \hat{\theta}_k)$;
> > > Receive $r_{k,t} = v_{k_t} \cdot \mathbb{1}_t$;
> > **end for**
> > Solve: $\hat{\theta}_{k+1} = \arg\min_{\theta \in \mathbb{H}} \hat{L}_k(\theta)$, where
> > $\hat{L}_k(\theta) = \frac{1}{\tau_k} \sum_{t=1}^{\tau_k} l_{k,t}(\theta)$.
> **end for**

**Algorithm 2** Online Newton Step Pricing (ONSP)

> **Input:** Convex and bounded set $\mathbb{H}$, $\theta_1$, parameter $\gamma, \epsilon > 0$
> Set $A_0 = \epsilon \cdot I_d$;
> **for** $t = 1$ **to** $T$ **do**
> > Observe $x_t$;
> > Set price $v_t = J(x_t^\top \theta_t)$;
> > Receive $r_t = v_t \cdot \mathbb{1}_t$;
> > Set surrogate loss function $l_t(\theta)$;
> > Calculate $\nabla_t = \nabla l_t(\theta)$;
> > Rank-1 update: $A_t = A_{t-1} + \nabla_t \nabla_t^\top$;
> > Newton step: $\hat{\theta}_{t+1} = \theta_t - \frac{1}{\gamma} A_t^{-1} \nabla_t$;
> > Projection: $\theta_{t+1} = \prod_{\mathbb{H}}^{A_t} (\hat{\theta}_{t+1})$.
> **end for**

### 4.1 Pricing with Distribution-Free Stochastic Features

**Assumption 4** (Stochastic features). *Assume $x_t \sim \mathbb{D} \subseteq D$ are independently identically distributed (i.i.d.) from an unknown distribution, for any $t = 1, 2, \ldots, T$.*

The first algorithm, Epoch-based Max-Likelihood Pricing (EMLP) algorithm, is suitable for a stochastic setting defined by Assumption 4. EMLP proceeds in epochs with each stage doubling the length of the previous epoch. At the end of each epoch, we consolidate the observed data and solve a maximum likelihood estimation problem to learn $\theta$. A max likelihood estimator (MLE) obtained by minimizing $\hat{L}_k(\theta) := \frac{1}{\tau_k} \sum_{t=1}^{\tau_k} l_{k,t}(\theta)$, which is then used in the next epoch as if it is the true parameter vector. In the equation, $k, \tau_k$ denotes the index and length of epoch $k$. The estimator is computed using data in $hist(k)$, which denotes the transcript for epoch $1 \sim k$. The pseudo-code of EMLP is summarized in Algorithm 1. In the remainder of this section, we discuss the computational efficiency and prove the upper regret bound of $O(d \log T)$.

**Computational Efficiency.** The calculations in EMLP are straightforward except for $\arg\min \hat{L}_k(\theta)$ and $J(u)$. As $g(v, u)$ is proved unimodal in Lemma 2, we may efficiently calculate $J(u)$ by binary search. We will prove that $l_{k,t}$ is exp-concave (and thus also convex). Therefore, we may apply any off-the-shelf tools for solving convex optimization.

**MLE and Probit Regression.** A closer inspection reveals that this log-likelihood function corresponds to a probit [Aldrich et al., 1984] or a logit model [Wright, 1995] for Gaussian or logistic noises. See Appendix C.2.1.

**Affine Invariance.** Both optimization problems involved depend only on $x^\top \theta$, so if we add any affine transformation to $x$ into $\tilde{x} = Ax$, the agent can instead learn a new parameter of $\tilde{\theta}^* = (A^\top)^{-1} \theta^*$ and achieve the same $u_t = x_t^\top \theta^*$. Also, the regret bound is not affected as the upper bound $B$ over $x^\top \theta$ does not change [6]. Therefore, it is only natural that the regret bound does not depend on the distribution $x$, nor the condition numbers of $\mathbb{E}[xx^\top]$ (i.e., the ratio of $\lambda_{\max}/\lambda_{\min}$).

### 4.2 Pricing with Adversarial Features

In this part, we propose an "Online Newton Step Pricing (ONSP)" algorithm that deals with adversarial $\{x_t\}$ series and guarantees $O(d \log T)$ regret. The pseudo-code of ONSP is shown as Algorithm 2. In each round, it uses the likelihood function as a surrogate loss and applies "Online Newton Step"(ONS) method to update $\hat{\theta}$. In the next round, it adopts the updated $\hat{\theta}$ and sets a price greedily. In the remainder of this section, we discuss some properties of ONSP and prove the regret bound.

The calculations of ONSP are straightforward. The time complexity of calculating the matrix inverse $A_t^{-1}$ is $O(d^3)$, which is fair as $d$ is small. In high-dimensional cases, we may use *Woodbury matrix identity*[7] to reduce it to $O(d^2)$ as we could get $A^{-1}$ directly from the latest round.

---

[6]Here $A$ is assumed invertible, otherwise the mapping from $\tilde{x}_t$ to $u_t$ does not necessarily exist.

[7]$(A + xx^\top)^{-1} = A^{-1} - \frac{1}{1+x^\top A^{-1} x} A^{-1} x (A^{-1} x)^\top$.

## 5 Regret Analysis

In this section, we mainly prove the logarithmic regret bounds of EMLP and ONSP corresponding to stochastic and adversarial settings, respectively. Besides, we also prove an $\Omega(\sqrt{T})$ regret bound on fully parametric $F$ with one parameter unknown.

### 5.1 $O(d\log T)$ Regret of EMLP

In this part, we present the regret analysis of Algorithm 1. First of all, we propose the following theorem as our main result on EMLP.

**Theorem 3** (Overall regret). *With Assumptions 1, 2, 3 and 4, the expected regret of EMLP can be bounded by:*

$$\mathbb{E}[Reg_{EMLP}] \leq 2C_s d\log T, \tag{6}$$

*where $C_s$ is a constant that depends only on $F(\omega)$ and is independent to $\mathbb{D}$.*

The proof of Theorem 3 is sophisticated. For the sake of clarity, we next present an inequality system as a roadmap toward the proof. After this, we formally illustrate each line of it with lemmas.

Since EMLP proposes $J(x_{k,t}^\top \hat{\theta}_k)$ in every round of epoch $k$, we may denote the per-round regret as $Reg_t(\hat{\theta}_k)$, where:

$$Reg_t(\theta) := g(J(x_t^\top \theta^*), x_t^\top \theta^*) - g(J(x_t^\top \theta), x_t^\top \theta^*). \tag{7}$$

Therefore, it is sufficient to prove the following Theorem:

**Theorem 4** (Expected per-round regret). *For the per-round regret defined in Equation (7), we have:*

$$\mathbb{E}[Reg_{k,t}(\hat{\theta}_k)] \leq C_s \cdot \frac{d}{\tau_k}.$$

The proof roadmap of Theorem 4 can be written as the following inequality system.

$$\mathbb{E}[Reg_{k,t}(\hat{\theta}_k)] \leq C \cdot \mathbb{E}[(\hat{\theta}_k - \theta^*)^\top x_{k,t} x_{k,t}^\top (\hat{\theta}_k - \theta^*)] \leq \frac{2C}{C_{\text{down}}} \mathbb{E}[\hat{L}_k(\hat{\theta}_k) - \hat{L}_k(\theta^*)] \leq \frac{2C \cdot C_{\exp}}{C_{\text{down}}^2} \frac{d}{\tau_k}. \tag{8}$$

We explain Equation (8) in details. The first inequality comes from the following Lemma 5.

**Lemma 5** (Quadratic regret bound). *We have:*

$$Reg_t(\theta) \leq C \cdot (\theta - \theta^*)^\top x_t x_t^\top (\theta - \theta^*), \forall \theta \in \mathbb{H}, \forall x_t \in \mathbb{D}. \tag{9}$$

*Here $C = 2B_f + (B + J(0)) \cdot B_{f'}$.*

The intuition is that function $g(J(u), u)$ is $2^{nd}$-order-smooth at $(J(u^*), u^*)$. A detailed proof of Lemma 5 is in Appendix B.2.1. Note that $C$ is highly dependent on the distribution $F$. After this, we propose Lemma 6 that contributes to the second inequality of Equation (8).

**Lemma 6** (Quadratic likelihood bound). *For the expected likelihood function $L_t(\theta)$ defined in Equation (2), we have:*

$$L_t(\theta) - L_t(\theta^*) \geq \frac{1}{2} C_{down} (\theta - \theta^*)^\top x_t x_t^\top (\theta - \theta^*), \forall \theta \in \mathbb{H}, \forall x \in \mathbb{D}, \tag{10}$$

$$where \quad C_{down} := \inf_{\omega \in [-B, B+J(0)]} \min \left\{ \frac{d^2 \log(1 - F(\omega))}{d\omega^2}, \frac{d^2 \log(F(\omega))}{d\omega^2} \right\} > 0. \tag{11}$$

*Proof.* Since the true parameter always maximizes the expected likelihood function [Murphy, 2012], by Taylor Expansion we have $\nabla L(\theta^*) = 0$, and hence $L_t(\theta) - L_t(\theta^*) = \frac{1}{2}(\theta - \theta^*)^\top \nabla^2 L_t(\tilde{\theta})(\theta - \theta^*)$ for some $\tilde{\theta} = \alpha\theta^* + (1 - \alpha)\theta$. Therefore, we only need to prove the following lemma:

**Lemma 7** (Strong convexity and Exponential Concavity). *Suppose $l_t(\theta)$ is the negative log-likelihood function in epoch $k$ at time $t$. For any $\theta \in \mathbb{H}, x_t \sim \mathbb{D}$, we have:*

$$\nabla^2 l_t(\theta) \succeq C_{down} x_t x_t^\top \succeq \frac{C_{down}}{C_{exp}} \nabla l_t(\theta) \nabla l_t(\theta)^\top \succeq 0, \tag{12}$$

$$where \ \ C_{exp} := \sup_{\omega \in [-B, B+J(0)]} \max \left\{ \frac{f(\omega)^2}{F(\omega)^2}, \frac{f(\omega)^2}{(1-F(\omega))^2} \right\} < +\infty. \tag{13}$$

Proof of Lemma 7 is in Appendix B.2.2. With this lemma, we see that Lemma 6 holds. ∎

With Lemma 5 and Lemma 6, we can immediately get the following Lemma 8.

**Lemma 8** (Surrogate Regret). *The relationship between $Reg(\theta)$ and likelihood function can be shown as follows:*

$$Reg_t(\theta) \leq \frac{2 \cdot C}{C_{down}} \left( L_t(\theta) - L_t(\theta^*) \right), \tag{14}$$

$\forall \theta \in \mathbb{H}, \forall x \in \mathbb{D}$, where $C$ and $C_{down}$ are defined in Lemma 5 and 6 respectively.

Lemma 8 enables us to choose the negative log-likelihood function as a surrogate loss. This is not only an important insight of EMLP regret analysis, but also the foundation of ONSP design.

The last inequality of Equation (8) comes from this lemma:

**Lemma 9** (Per-epoch surrogate regret bound). *Denoting $\hat{\theta}_k$ as the estimator coming from epoch $(k-1)$ and being used in epoch $k$, we have:*

$$\mathbb{E}_h[\hat{L}_k(\hat{\theta}_k) - \hat{L}_k(\theta^*)] \leq \frac{C_{exp}}{C_{down}} \cdot \frac{d}{\tau_k + 1}. \tag{15}$$

*Here $C_{exp}$ is defined in Equation 13, and $\mathbb{E}_h[\cdot] = \mathbb{E}[\cdot | hist(k-1)]$.*

Proof of Lemma 9 is partly derived from the work Koren and Levy [2015], and here we give a proof sketch without specific derivations. A detailed proof lies in Appendix B.2.3.

*Proof sketch* of Lemma 9. We list the four main points that contribute to the proof:

- Notice that $l_{k,t}(\theta)$ is strongly convex w.r.t. a seminorm $x_{k,t} x_{k,t}^\top$, we know $\hat{L}_k(\theta)$ is also strongly convex w.r.t. $\sum_{t=1}^{\tau_k} x_{k,t} x_{k,t}^\top$.

- For two strongly convex functions $g_1$ and $g_2$, we can upper bound the distance between their arg-minimals (scaled by some norm $||\cdot||$) with the dual norm of $\nabla(g_1 - g_2)$.

- Since a seminorm has no dual norm, we apply two methods to convert it into a norm: (1) separation of parameters and likelihood functions with a "leave-one-out" method (to separately take expectations), and (2) separation of the spinning space and the null space.

- As the dual data-dependent norm offsets the sum of $xx^\top$ to a constant, Lemma 9 holds.

We have so far proved Inequality (8) after proving Lemma 5, 6, 9. Therefore, Theorem 4 holds.

## 5.2  $O(d \log T)$ **Regret of ONSP**

Here we present the regret analysis of Algorithm 2 (ONSP). Firstly, we state the main theorem.

**Theorem 10.** *With Assumptions 1, 2, 3, the regret of Algorithm 2 (ONSP) satisfies:*

$$Reg_{ONSP} \leq C_a \cdot d \log T, \tag{16}$$

*where $C_a$ is a function only dependent on $F$.*

*Proof.* Proof of Theorem 10 here is more concise than Section 5.1, because the important Lemma 7 and 8 have been proved there. From Lemma 8, we have:

$$g(J(u_t^*), u_t^*) - g(J(u_t), u_t^*) \leq \frac{2 \cdot C}{C_{\text{down}}} \cdot \mathbb{E}_{N_t}[l_t(\theta_t) - l_t(\theta^*)]. \tag{17}$$

With Equation 17, we may reduce the regret of likelihood functions as a surrogate regret of pricing. From Lemma 7 we see that the log-likelihood function is $\frac{C_{\text{down}}}{C_{\text{exp}}}$-exponentially concave[8]. This enables

---

[8]A function $f(\mu)$ is $\alpha$-exponentially concave iff $\nabla^2 f(\mu) \succeq \alpha \nabla f(\mu) \nabla f(\mu)^\top$.

an application of Online Newton Step method to achieve a logarithmic regret. Therefore, by citing from the *Online Convex Optimization* [Hazan, 2016], we have the following Lemma.

**Lemma 11** (Online Newton Step). *With parameters $\gamma = \frac{1}{2}\min\{\frac{1}{4GD}, \alpha\}$ and $\epsilon = \frac{1}{\gamma^2 D^2}$, and $T > 4$ guarantees:*

$$\sup_{\{x_t\}}\left\{\sum_{t=1}^{T} l_t(\theta_t) - \min_{\theta \in \mathbb{H}}\sum_{t=1}^{T} l_t(\theta)\right\} \leq 5\left(\frac{1}{\alpha} + GD\right) d\log T.$$

*Here $\alpha = \frac{C_{down}}{C_{exp}}$, $D = 2 \cdot B_1$ and $G = \sqrt{C_{exp}} \cdot B_2$.*

With Equation 17 and Lemma 11, we have:

$$Reg = \sum_{t=1}^{T}\big(g(J(u_t^*), u_t^*) - \mathbb{E}_{N_1, N_2, \ldots, N_{t-1}}[g(J(u_t), u_t^*)]\big) \leq \frac{2 \cdot C}{C_{\text{down}}} \cdot 5\left(\frac{1}{\alpha} + GD\right) d\log T. \quad (18)$$

Therefore, we have proved Lemma 10. ∎

### 5.3 Lower Bound for Unknown Distribution

In this part, we evaluate Assumption 1 and prove that an $\Omega(\sqrt{T})$ lower regret bound is unavoidable with even a slight relaxation: a Gaussian noise with unknown $\sigma$. Our proof is inspired by Broder and Rusmevichientong [2012] Theorem 3.1, while our lower bound relies on more specific assumptions (and thus applies to more general cases).

We firstly state Assumption 5 covering this part, and then state Theorem 12 as a lower bound:

**Assumption 5.** *The noise $N_t \sim \mathcal{N}(0, \sigma^2)$ independently, where $0 < \sigma \leq 1$ is fixed and **unknown**.*

**Theorem 12** (Lower bound with unknown $\sigma$). *Under Assumption 2, 3, 4 and 5, for any policy (algorithm) $\Psi : \mathbb{R}^d \times (\mathbb{R}^d, \mathbb{R}, \{0, 1\})^{t-1} \to \mathbb{R}^+$ and any $T > 2$, there exists a Gaussian parameter $\sigma \in \mathbb{R}^+$, a distribution $\mathbb{D}$ of features and a fixed parameter $\theta^*$, such that: $Reg_{\Psi} \geq \frac{1}{24000} \cdot \sqrt{T}$.*

*Remark:* Here we assume $x_t$ to be i.i.d., which also implies the applicability on adversarial features. However, the minimax regret of the stochastic feature setting is $\Theta(\sqrt{T})$ [Javanmard and Nazerzadeh, 2019], while existing results have not yet closed the gap in adversarial feature settings.

*Proof sketch* of Theorem 12. Here we assume a fixed valuation, i.e. $u^* = x_t^\top \theta^*, \forall t = 1, 2, \ldots$. Equivalently, we assume a fixed feature. The main idea of proof is similar to that in Broder and Rusmevichientong [2012]: we assume $\sigma_1 = 1, \sigma_2 = 1 - T^{-\frac{1}{4}}$, and we prove that: (1) it is costly for an algorithm to perform well in both cases if the $\sigma$'s are different by a lot, and (2) it is costly for an algorithm to distinguish the two cases if $\sigma$'s are close enough to each other. We put the detailed proof in Appendix B.3.

## 6 Numerical Result

In this section, we conduct numerical experiments to validate EMLP and ONSP. In comparison with the existing work, we implement a discretized EXP-4 [Auer et al., 2002] algorithm for pricing, as is introduced in Cohen et al. [2020] (in a slightly different setting). We will test these three algorithms in both stochastic and adversarial settings.

Basically, we assume $d = 2, B_1 = B_2 = B = 1$ and $N_t \sim \mathcal{N}(0, \sigma^2)$ with $\sigma = 0.25$. In both settings, we conduct EMLP and ONSP for $T = 2^{16}$ rounds. For ONSP, we empirically select $\gamma$ and $\epsilon$ that accelerates the convergence, instead of using the values specified in Lemma 11. Since EXP-4 consumes exponential time and requires the knowledge of $T$ in advance to discretize the policy and valuation spaces, we execute EXP-4 for a series of $T = 2^k, k = 1, 2, \ldots, 12$. We repeat every experiment 5 times for each setting and then take an average.

**Stochastic Setting.** We implement and test EMLP, ONSP and EXP-4 with stochastic $\{x_t\}$'s. The numerical results are shown in Figure 1a on a log-log diagram, with the regrets divided by $\log(t)$. It shows $\log(t)$-convergences on EMLP and ONSP, while EXP-4 is in a $t^\alpha$ rate with $\alpha \approx 0.699$.

**Adversarial Setting.** We implement the three algorithms and test them with an adversarial $\{x_t\}$'s: for the $k$-th epoch, i.e. $t = 2^{k-1}, 2^{k-1} + 1, \ldots, 2^k - 1$, we let $x_t = [1, 0]^\top$ if $k \equiv 1 \pmod 2$ and

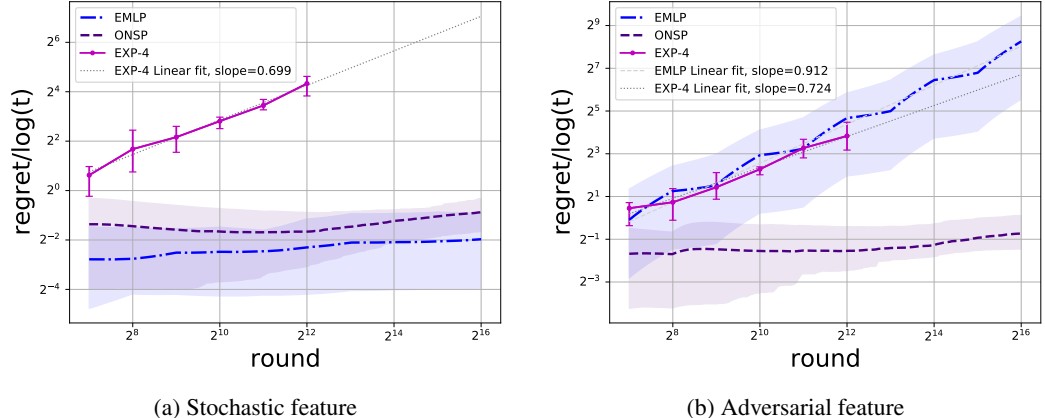

(a) Stochastic feature

(b) Adversarial feature

Figure 1: The regret of EMLP, ONSP and EXP-4 on simulated examples (we only conduct EXP-4 up to $T = 2^{12}$ due to its exponential time consuming), with Figure a for stochastic features and Figure b for adversarial ones. The plots are in log-log scales with all regrets divided by a $\log(t)$ factor to show the convergence. For EXP-4, we discretize the parameter space with $T^{-\frac{1}{3}}$-size grids, which would incur an $\tilde{O}(T^{\frac{2}{3}})$ regret according to Cohen et al. [2020]. We also plot linear fits for some regret curves, where a slope-$\alpha$ line indicates an $O(T^{\alpha})$ regret. Besides, we draw error bars and bands with 0.95 coverage using Wald's test. The two diagrams reveal that (i) logarithmic regrets of EMLP and ONSP in the stochastic setting, (ii) a nearly-linear regret of EMLP in the adversarial setting, and (iii) $O(T^{\frac{2}{3}})$ regrets of EXP-4 in both settings.

$x_t = [0, 1]^{\top}$ if $k \equiv 0(\mod 2)$. The numerical results are shown in Figure 1b on a log-log diagram, with the regrets divided by $\log(t)$. The log-log plots of ONSP and EXP-4 are almost the same as those in Figure 1a. However, EMLP shows an almost linear ($t^{\alpha}$ rate with $\alpha \approx 0.912$) regret in this adversarial setting. This is because the adversarial series only trains one dimension of $\theta$ in each epoch, while the other is arbitrarily initialized and does not necessarily converge. However, in the next epoch, the incorrect dimension is exploited. Therefore, a linear regret originates.

## 7 Discussion

Here we discuss the coefficients on our regret bounds as a potential extension of future works. In Appendix C we will discuss more on algorithmic design, problem modeling, and ethic issues.

**Coefficients on Regret Bounds.** The exact regret bounds of both EMLP and ONSP contain a constant $\frac{C_{\exp}}{C_{\mathrm{down}}}$ that highly depends on the noise CDF $F$ and could be large. A detailed analysis in Appendix C.1 shows that $\frac{C_{\exp}}{C_{\mathrm{down}}}$ is exponentially large w.r.t. $\frac{B}{\sigma}$ (see Equation 39 and Lemma 21) for Gaussian noise $\mathcal{N}(0, \sigma^2)$, which implies that a smaller noise variance would lead to a (much) larger regret bound. This is very counter-intuitive as a larger noise usually leads to a more sophisticated situation, but similar phenomenons also occur in existing algorithms that are suitable for constant-variance noise, such as RMLP in Javanmard and Nazerzadeh [2019] and OORMLP in Wang et al. [2020]. In fact, it is because a (constantly) large noise would help explore the unknown parameter $\theta^*$ and smoothen the expected regret. In this work, this can be addressed by increasing $T$ since we mainly concern the asymptotic regrets as $T \to \infty$ with fixed noise distributions. However, we admit that it is indeed a nontrivial issue for finite $T$ and small $\sigma$ situations. There exists a "ShallowPricing" method in Cohen et al. [2020] that can deal with a very-small-variance noise setting (when $\sigma = \tilde{O}(\frac{1}{T})$) and achieve a logarithmic regret. Specifically, its regret bound would decrease as the noise variance $\sigma$ decreases (but would still not reach $O(\log \log T)$ as the noise vanishes). We might also apply this method as a preprocess to cut the parameter domain and decrease $\frac{B}{\sigma}$ within logarithmic trials (see Cohen et al. [2020] Thm. 3), but it is still open whether a $\log(T)$ regret is achievable when $\sigma = \Theta(T^{-\alpha})$ for $\alpha \in (0, 1)$.

## 8 Conclusion

In this work, we studied the problem of online feature-based dynamic pricing with a noisy linear valuation in both stochastic and adversarial settings. We proposed a max-likelihood-estimate-based algorithm (EMLP) for stochastic features and an online-Newton-step-based algorithm (ONSP) for

adversarial features. Both of them enjoy a regret guarantee of $O(d \log T)$, which also attains the information-theoretic limit up to a constant factor. Compared with existing works, EMLP gets rid of strong assumptions on the distribution of the feature vectors in the stochastic setting, and ONSP improves the regret bound exponentially from $O(T^{2/3})$ to $O(\log T)$ in the adversarial setting. We also showed that knowing the noise distribution (or the demand curve) is required to obtain logarithmic regret, where we prove a lower bound of $\Omega(\sqrt{T})$ on the regret for the case when the noise is knowingly Gaussian but with an unknown $\sigma$. In addition, we conducted numerical experiments to empirically validate the scaling of our algorithms. Finally, we discussed the regret dependence on the noise variance, and proposed a subtle open problem for further study.

## Acknowledgments

The work is partially supported by the Adobe Data Science Award and a start-up grant from the UCSB Department of Computer Science. We appreciate the input from anonymous reviewers and AC as well as a discussion with Akshay Krishnamurthy for clarifying some details of Krishnamurthy et al. [2021].

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
