# APPENDIX

## A  Other related works

Here we will briefly review the history and recent studies that are related to our work. For the historical introductions, we mainly refer to den Boer [2015] as a survey. For bandit approaches, we will review some works that apply bandit algorithms to settle pricing problems. For the structural models, we will introduce different modules based on the review in Chan et al. [2009]. Based on the existing works, we might have a better view of our problem setting and methodology.

### A.1  History of Pricing

It was the work of Cournot [1897] in 1897 that firstly applied mathematics to analyze the relationship between prices and demands. In that work, the price was denoted as $p$ and the demand was defined as a *demand function* $F(p)$. Therefore, the revenue could be written as $pF(p)$. This was a straightforward interpretation of the general pricing problem, and the key to solving it was estimations of $F(p)$ regarding different products. Later in 1938, the work Schultz et al. [1938] proposed price-demand measurements on exclusive kinds of products. It is worth mentioning that these problems are "static pricing" ones, because $F$ is totally determined by price $p$ and we only need to insist on the optimal one to maximize our profits.

However, the static settings were qualified by the following two observations: on the one hand, a demand function may not only depends on the static value of $p$, but also be affected by the trend of $p$'s changing [Evans, 1924, Mazumdar et al., 2005]; on the other hand, even if $F(p)$ is static, $p$ itself might change over time according to other factors such as inventory level [Kincaid and Darling, 1963]. As a result, it is necessary to consider dynamics in both demand and price, which leads to a "dynamic pricing" problem setting.

### A.2  Dynamic Pricing as Bandits

As is said in Section 2, the pricing problem can be viewed as a stochastic contextual bandits problem [see, e.g., Langford and Zhang, 2007, Agarwal et al., 2014]. Even though we may not know the form of the demand function, we can definitely see feedback of demands, i.e. how many products are sold out, which enables us to learn a better decision-making policy. Therefore, it can be studied in a bandit module. If the demand function is totally agnostic, i.e. the evaluations (the highest prices that customers would accept) come at random or even at adversary over time, then it can be modeled as a Multi-arm bandit (MAB) problem [Whittle, 1980] exactly. In our paper, instead, we focus on selling different products with a great variety of features. This can be characterized as a Contextual bandit (CB) problem [Auer et al., 2002, Langford and Zhang, 2007]. The work Cohen et al. [2020], which applies the "EXP-4" algorithm from Auer et al. [2002], also mentions that "the arms represent prices and the payoffs from the different arms are correlated since the measures of demand evaluated at different price points are correlated random variables". A variety of existing works, including Kleinberg and Leighton [2003], Araman and Caldentey [2009], Chen and Farias [2013], Keskin and Zeevi [2014], Besbes and Zeevi [2015], has been approaching the demand function from a perspective of from either parameterized or non-parameterized bandits.

However, our problem setting is different from a contextual bandits setting in at least two perspectives: feedback and regret. The pricing problem has a specially structured feedback between full information and bandits setting. Specifically, $r_t > 0$ implies that all policies producing $v < v_t$ will end up receiving $r'_t = v$, and $r_t = 0$ implies that all policies producing $v > v_t$ will end up receiving $r'_t = 0$. However, the missing patterns are confounded with the rewards. Therefore it is non-trivial to leverage this structure to improve the importance sampling approach underlying the algorithm of Agarwal et al. [2014]. We instead consider the natural analog to the linear contextual bandits setting [Chu et al., 2011][9] and demonstrate that in this case an exponential improvement in the regret is possible using the additional information from the censored feedback. As for regret, while in contextual bandits it refers to a comparison with the optimal policy, it is here referring to a comparison with the optimal *action*. In other words, though our approaches (both in EMLP and in ONSP) are finding the true parameter $\theta^*$, the regret is defined as the "revenue gap" between the optimal price and our proposed prices. These are actually equivalent in our fully-parametric setting (where we assume a

---

[9]But do notice that our expected reward above is not linear, even if the valuation function is.

linear-valuation-known-noise model), but will differ a lot in partially parametric and totally agnostic settings.

## A.3 Structural Model

While a totally agnostic model guarantees the most generality, a structural model would help us better understand the mechanism behind the observation of prices and demands. The key to a structural pricing model is the *behavior* of agents in the market, including customers and/or firms. In other words, the behavior of each side can be described as a decision model. From the perspective of demand (customers), the work Kadiyali et al. [1996] adopts a linear model on laundry detergents market, Iyengar et al. [2007] and Lambrecht et al. [2007] study three-part-tariff pricing problems on wireless and internet services with mixed logit models. Besanko et al. assumed an aggregate logit model on customers in works Besanko et al. [1998] and Besanko et al. [2003] in order to study the competitive behavior of manufacturers in ketchup market. Meanwhile, the supply side is usually assumed to be more strategic, such as Bertrand-Nash behaviors [Kadiyali et al., 1996, Besanko et al., 1998, Draganska and Jain, 2006]. For more details, please see Chan et al. [2009].

# B  Proofs

## B.1  Proof of Lemma 2

*Proof.* Since $v^* = \arg\max g(v, u)$, we have:

$$
\begin{aligned}
&\frac{\partial g(v, u)}{\partial v}\Big|_{v=v^*} = 0 \\
\Leftrightarrow\, &1 - F(v^* - u) - v^* \cdot f(v^* - u) = 0 \\
\Leftrightarrow\, &\frac{1 - F(v^* - u)}{f(v^* - u)} - (v^* - u) = u
\end{aligned}
$$

Define $\varphi(\omega) = \frac{1-F(\omega)}{f(\omega)} - \omega$, and we take derivatives:

$$
\varphi'(\omega) = \frac{-f^2(\omega) - (1 - F(\omega))f'(w)}{f^2(w)} - 1 = \frac{\mathrm{d}^2 \log(1 - F(\omega))}{\mathrm{d}\omega^2} \cdot \frac{(1 - F(\omega))^2}{(f(\omega))^2} - 1 < -1,
$$

where the last equality comes from the strict log-concavity of $(1 - F(\omega))$. Therefore, $\varphi(\omega)$ is decreasing and $\varphi(+\infty) = -\infty$. Also, notice $\varphi(-\infty) = +\infty$, we know that for any $u \in \mathbb{R}$, there exists an $\omega$ such that $\varphi(\omega) = u$. For $u \geq 0$, we know that $g(v, u) \geq 0$ for $v \geq 0$ and $g(v, u) < 0$ for $v < 0$. Therefore, $v^* \geq 0$ if $u \geq 0$. ∎

## B.2  Proofs in Section 5.1

### B.2.1  Proof of Lemma 5

*Proof.* We again define $\varphi(\omega) = \frac{1-F(\omega)}{f(\omega)} - \omega$ as in Appendix B.1. According to Equation 5, we have:

$$
\begin{aligned}
&\frac{\partial g(v, u)}{\partial v}\Big|_{v=J(u)} = 0 \\
\Rightarrow\, &1 - F(J(u) - u) - J(u) \cdot f(J(u) - u) = 0 \\
\Rightarrow\, &\varphi(J(u) - u) = u \\
\Rightarrow\, &J(u) = u + \varphi^{-1}(u) \\
\Rightarrow\, &J'(u) = 1 + \frac{1}{\varphi'(\varphi^{-1}(u))}.
\end{aligned} \tag{19}
$$

The last line of Equation 19 is due to the Implicit Function Derivatives Principle. From the result in Appendix B.1, we know that $\varphi'(\omega) < -1, \forall \omega \in \mathbb{R}$. Therefore, we have $J'(u) \in (0, 1), u \in \mathbb{R}$, and hence $J(0) \leq J(u) < u + J(0)$ for $u \geq 0$ and $u + J(0) \leq J(u) \leq J(0)$ for $u \leq 0$. Since $u \in [-B, B]$, we may assume that $v \in [0, B + J(0)]$ without losing generality. In the following part, we will frequently use this range.

Denote $u := x_t^\top \theta, u^* = x_t^\top \theta^*$. According to Equation 7, we know that:

$$
\begin{aligned}
Reg_t(\theta) &= g(J(u^*), u^*) - g(J(u), u^*) \\
&= -\frac{\partial g(v, u^*)}{\partial v}\Big|_{v=J(u^*)}(J(u^* - J(u))) + \frac{1}{2}\left(-\frac{\partial^2 g(v, u^*)}{\partial v^2}\Big|_{v=\tilde{v}}\right)(J(u^*) - J(u))^2 \\
&\leq 0 + \frac{1}{2}\max_{\tilde{v}\in[0,B+J(0)]}\left(-\frac{\partial^2 g(v, u^*)}{\partial v^2}\Big|_{v=\tilde{v}}\right)\cdot(J(u^*) - J(u))^2 \\
&= \frac{1}{2}\max_{\tilde{v}\in[0,B+J(0)]}(2f(\tilde{v} - u^*) + \tilde{v}\cdot f'(\tilde{v} - u^*))\cdot(J(u^*) - J(u))^2 \\
&\leq \frac{1}{2}(2B_f + (B + J(0))\cdot B_{f'})(J(u^*) - J(u))^2 \\
&\leq \frac{1}{2}(2B_f + (B + J(0))\cdot B_{f'})(u^* - u)^2 \\
&= \frac{1}{2}(2B_f + (B + J(0))\cdot B_{f'})(\theta^* - \theta)^\top x_t x_t^\top(\theta^* - \theta).
\end{aligned}
$$

Here the first line is from the definition of $g$ and $Reg(\theta)$, the second line is due to Taylor's Expansion, the third line is from the fact that $J(u^*)$ maximizes $g(v, u^*)$ with respect to $v$, the fourth line is by calculus, the fifth line is from the assumption that $0 < f(\omega) \leq B_f, |f'(\omega)| \leq B_{f'}$ and $v \in [0, B + J(0)]$, the sixth line is because of $J'(u) \in (0, 1), \forall u \in \mathbb{R}$, and the seventh line is from the definition of $u$ and $u^*$. ∎

### B.2.2 Proof of Lemma 7

*Proof.* We take derivatives of $l_t(\theta)$, and we get:

$$
\begin{aligned}
l_t(\theta) =& \mathbb{1}_t\left(-\log(1 - F(v_t - x_t^\top\theta))\right) + (1 - \mathbb{1}_t)\left(-\log(F(v_t - x_t^\top\theta))\right) \\
\nabla l_t(\theta) =& \mathbb{1}_t\left(-\frac{f(v_t - x_t^\top\theta)}{1 - F(v_t - x_t^\top\theta)}\right)\cdot x_t + (1 - \mathbb{1}_t)\left(\frac{f(v_t - x_t^\top\theta)}{F(v_t - x_t^\top\theta)}\right)\cdot x_t \\
\nabla^2 l_t(\theta) =& \mathbb{1}_t\cdot\frac{f(v_t - x_t^\top\theta)^2 + f'(v_t - x_t^\top\theta)\cdot(1 - F(v_t - x_t^\top\theta))}{(1 - F(v_t - x_t^\top\theta))^2}\cdot x_t x_t^\top \\
&+ (1 - \mathbb{1}_t)\cdot\frac{f(v_t - x_t^\top\theta)^2 - f'(v_t - x_t^\top\theta)F(v_t - x_t^\top\theta)}{F(v_t - x_t^\top\theta)^2}\cdot x_t x_t^\top \\
=& \mathbb{1}_t\cdot\frac{-\mathrm{d}^2\log(1 - F(\omega))}{\mathrm{d}\omega^2}\Big|_{\omega=v_t-x_t^\top\theta}\cdot x_t x_t^\top + (1 - \mathbb{1}_t)\frac{-\mathrm{d}^2\log(F(\omega))}{\mathrm{d}\omega^2}\Big|_{\omega=v_t-x_t^\top\theta}\cdot x_t x_t^\top \\
\succeq& \inf_{\omega\in[-B,B+J(0)]}\min\left\{\frac{\mathrm{d}^2\log(1 - F(\omega))}{\mathrm{d}\omega^2}, \frac{\mathrm{d}^2\log(F(\omega))}{\mathrm{d}\omega^2}\right\} \\
=& C_{\text{down}}x_t x_t^\top,
\end{aligned}
$$

(20)

which directly proves the first inequality. For the second inequality, just notice that

$$
\begin{aligned}
\nabla l_t(\theta)\nabla l_t(\theta)^\top =& \mathbb{1}_t\left(\frac{f(v_t - x_t^\top\theta)}{1 - F(v_t - x_t^\top\theta)}\right)^2 x_t x_t^\top + (1 - \mathbb{1}_t)\left(\frac{f(v_t - x_t^\top\theta)}{F(v_t - x_t^\top\theta)}\right)^2 x_t x_t^\top \\
\preceq& \sup_{\omega\in[-B,B+J(0)]}\max\left\{\left(\frac{f(\omega)}{F(\omega)}\right)^2, \left(\frac{f(\omega)}{1 - F(\omega)}\right)^2\right\}x_t x_t^\top \\
=& C_{\exp}x_t x_t^\top.
\end{aligned}
$$

(21)

The only thing to point out is that $\frac{f(\omega)}{F(\omega)}$ and $\frac{f(\omega)}{1-F(\omega)}$ are all continuous for $\omega \in [-B, B + J(0)]$, as $F(\omega)$ is strictly increasing and thus $0 < F(\omega) < 1, \omega \in \mathbb{R}$. ∎

### B.2.3 Proof of Lemma 9

*Proof.* In the following part, we consider a situation that an epoch of $n \geq 2$ rounds of pricing is conducted, generating $l_j(\theta)$ as negative likelihood functions, $j = 1, 2, \ldots, n$. Define a "**leave-one-**

**out**"negative log-likelihood function

$$\tilde{L}_i(\theta) = \frac{1}{n} \sum_{j=1, j \neq i}^{n} l_j(\theta),$$

and let

$$\tilde{\theta}_i := \arg\min_{\theta} \tilde{L}_i(\theta).$$

Based on this definition, we know that $\tilde{\theta}_i$ is independent to $l_i(\theta)$ given historical data, and that $\tilde{\theta}_i$ are identically distributed for all $i = 1, 2, 3, \ldots, n$.

In the following part, we will firstly propose and proof the following inequality:

$$\frac{1}{n} \sum_{i=1}^{n} (l_i(\tilde{\theta}_i) - l_i(\hat{\theta})) \leq \frac{C_{\exp}}{C_{down}} \frac{d}{n} = O(\frac{d}{n}), \tag{22}$$

where $\hat{\theta}$ is the short-hand notation of $\hat{\theta}_k$ as we do not specify the epoch $k$ in this part. We now cite a lemma from Koren and Levy [2015]:

**Lemma 13.** *Let $g_1$, $g_2$ be 2 convex function defined over a closed and convex domain $\mathcal{K} \subseteq \mathbb{R}^d$, and let $x_1 = \arg\min_{x \in \mathcal{K}} g_1(x)$ and $x_2 = \arg\min_{x \in \mathcal{K}} g_2(x)$. Assume $g_2$ is locally $\delta$-strongly-convex at $x_1$ with respect to a norm $|| \cdot ||$. Then, for $h = g_2 - g_1$ we have*

$$||x_2 - x_1|| \leq \frac{2}{\delta} ||\nabla h(x_1)||_*.$$

*Here $|| \cdot ||_*$ denotes a dual norm.*

The following is a proof of this lemma.

*Proof.* (of Lemma 13) According to convexity of $g_2$, we have:

$$g_2(x_1) \geq g_2(x_2) + \nabla g_2(x_2)^\top (x_1 - x_2). \tag{23}$$

According to strong convexity of $g_2$ at $x_1$, we have:

$$g_2(x_2) \geq g_2(x_1) + \nabla g_2(x_1)^\top (x_2 - x_1) + \frac{\delta}{2} ||x_2 - x_1||^2. \tag{24}$$

Add Equation (23) and (24), and we have:

$$g_2(x_1) + g_2(x_2) \geq g_2(x_2) + g_2(x_1) + (\nabla g_2(x_1) - \nabla g_2(x_2))^\top (x_2 - x_1) + \frac{\delta}{2} ||x_2 - x_1||^2$$

$$\Leftrightarrow \quad (\nabla g_2(x_1) - \nabla g_2(x_2))^\top (x_1 - x_2) \geq \frac{\delta}{2} ||x_1 - x_2||^2$$

$$\Leftrightarrow \quad (\nabla g_1(x_1) + \nabla h(x_1) - \nabla g_2(x_2))^\top (x_1 - x_2) \geq \frac{\delta}{2} ||x_1 - x_2||^2$$

$$\Leftrightarrow \quad \nabla h(x_1)^\top (x_1 - x_2) \geq \frac{\delta}{2} ||x_1 - x_2||^2$$

$$\Rightarrow \quad ||\nabla h(x_1)||_* ||x_1 - x_2|| \geq \frac{\delta}{2} ||x_1 - x_2||^2$$

$$\Rightarrow \quad ||\nabla h(x_1)||_* \geq \frac{\delta}{2} ||x_1 - x_2||.$$

$$\tag{25}$$

The first step is trivial. The second step is a sequence of $g_2 = g_1 + h$. The third step is derived by the following 2 first-order optimality conditions: $\nabla g_1(x_1)^\top (x_1 - x_2) \leq 0$, and $\nabla g_2(x_2)^\top (x_2 - x_1) \leq 0$. The fourth step is derived from Holder's Inequality:

$$||\nabla h(x_1)||_* ||x_1 - x_2|| \geq \nabla h(x_1)^\top (x_1 - x_2).$$

Therefore, the lemma holds. ∎

In the following part, we will set up a strongly convex function of $g_2$. Denote $H = \sum_{t=1}^{n} x_t x_t^\top$. From Lemma 7, we know that

$$\nabla^2 \hat{L}(\theta) \succeq C_{down} \frac{1}{n} H.$$

Here $\hat{L}(\theta)$ is the short-hand notation of $\hat{L}_k(\theta)$ as we do not specify $k$ in this part. Since we do not know if $H$ is invertible, i.e. if a norm can be induced by $H$, we cannot let $g_2(\theta) = \hat{L}(\theta)$. Instead, we change the variable as follows:

We first apply singular value decomposition to $H$, i.e. $H = U\Sigma U^\top$, where $U \in \mathbb{R}^{d \times r}, U^\top U = I_r, \Sigma = diag\{\lambda_1, \lambda_2, \ldots, \lambda_r\} \succ 0$. After that, we introduce a new variable $\eta := U^\top \theta$. Therefore, we have $\theta = U\eta + V\epsilon$, where $V \in \mathbb{R}^{d \times (d-r)}, V^\top V = I_{d-r}, V^\top U = 0$ is the standard orthogonal bases of the null space of $U$, and $\epsilon \in \mathbb{R}^{(d-r)}$. Similarly, we define $\tilde{\eta}_i = U^\top \tilde{\theta}_i$ and $\hat{\eta} = U^\top \hat{\theta}$. According to these, we define the following functions:

$$\begin{aligned}
f_i(\eta) &:= l_i(\theta) = l_i(U\eta + V\epsilon) \\
\tilde{F}_i(\eta) &:= \tilde{L}_i(\theta) = \tilde{L}_i(U\eta + V\epsilon) \\
\hat{F}(\eta) &:= \hat{L}(\theta) = \hat{L}(U\eta + V\epsilon).
\end{aligned} \tag{26}$$

Now we prove that $\hat{F}(\eta)$ is locally-strongly-convex. Similar to the proof of Lemma 7, we have:

$$\begin{aligned}
\nabla^2 \hat{F}(\eta) &= \frac{1}{n} \sum_{i=1}^{n} \nabla^2 f_i(\eta) \\
&= \frac{1}{n} \sum_{i=1}^{n} \frac{\partial^2 l_i}{\partial(x_i^\top \theta)^2} \left(\frac{\partial x_i^\top \theta}{\partial \eta}\right) \left(\frac{\partial x_i^\top \theta}{\partial \eta}\right)^\top \\
&= \frac{1}{n} \sum_{i=1}^{n} \frac{\partial^2 l_i}{\partial(x_i^\top \theta)^2} \left(\frac{\partial x_i^\top (U\eta + V\epsilon)}{\partial \eta}\right) \left(\frac{\partial x_i^\top (U\eta + V\epsilon)}{\partial \eta}\right)^\top \\
&= \frac{1}{n} \sum_{i=1}^{n} \frac{\partial^2 l_i}{\partial(x_i^\top \theta)^2} (U^\top x_i)(U^\top x_i)^\top \\
&\succeq \frac{1}{n} \sum_{i=1}^{n} C_{down} U^\top x_i x_i^\top U \\
&= \frac{1}{n} C_{down} U^\top \left(\sum_{i=1}^{n} x_i x_i^\top\right) U^\top \\
&= \frac{1}{n} C_{down} U^\top H U \\
&= \frac{1}{n} C_{down} U^\top U \Sigma U^\top U \\
&= \frac{1}{n} C_{down} \Sigma \\
&\succ 0
\end{aligned} \tag{27}$$

That is to say, $\hat{F}(\eta)$ is locally $\frac{C_{down}}{n}$-strongly convex w.r.t $\Sigma$ at $\eta$. Similarly, we can verify that $\tilde{F}_i(\eta)$ is convex (not necessarily strongly convex). Therefore, according to Lemma 13, let $g_1(\eta) = \tilde{F}_i(\eta), g_2(\eta) = \hat{F}(\eta)$, and then $x_1 = \tilde{\eta}_i = U^\top \tilde{\theta}_i, x_2 = \hat{\eta} = U^\top \hat{\theta}$. Therefore, we have:

$$||\hat{\eta} - \tilde{\eta}_i||_\Sigma \leq \frac{1}{C_{down}} ||\nabla f_i(\tilde{\eta}_i)||_\Sigma^*. \tag{28}$$

Now let us show the validation of this theorem:

$$
\begin{aligned}
l_i(\tilde{\theta}_i) - l_i(\hat{\theta}) =& f_i(\tilde{\eta}_i) - f_i(\hat{\eta}) \\
\leq& \nabla f_i(\tilde{\eta}_i)^\top (\tilde{\eta}_i - \hat{\eta}) \\
&\underset{\substack{\uparrow \\ \text{convexity}}}{} \\
\leq& ||\nabla f_i(\tilde{\eta}_i)||_\Sigma^* ||\tilde{\eta}_i - \hat{\eta}||_\Sigma \\
&\underset{\substack{\uparrow \\ \text{Holder inequality}}}{} \\
\leq& \frac{1}{C_{down}} (||\nabla f_i(\tilde{\eta}_i)||_\Sigma^*)^2 . \\
&\underset{\substack{\uparrow \\ \text{Lemma 13}}}{}
\end{aligned}
\tag{29}
$$

And thus we have

$$
\begin{aligned}
\sum_{i=1}^n l_i(\tilde{\theta}_i) - l_i(\hat{\theta}) \leq& \frac{1}{C_{down}} \sum_{i=1}^n ||\nabla f_i(\tilde{\eta}_i)||_\Sigma^*{}^2 \\
\leq& \frac{1}{C_{down}} \sum_{i=1}^n (\frac{p}{\Phi})^2_{\max} x_i^\top U \Sigma^{-1} U^\top x_i \\
=& \frac{C_{\exp}}{C_{down}} C_{\exp} \sum_{i=1}^n tr(x_i^\top U \Sigma^{-1} U^\top x_i) \\
=& \frac{C_{\exp}}{C_{down}} C_{\exp} \sum_{i=1}^n tr(U \Sigma^{-1} U^\top x_i x_i^\top) \\
=& \frac{C_{\exp}}{C_{down}} C_{\exp} tr(U \Sigma^{-1} U^\top \sum_{i=1}^n x_i x_i^\top) \\
=& \frac{C_{\exp}}{C_{down}} C_{\exp} tr(U \Sigma^{-1} U^\top H) \\
=& \frac{C_{\exp}}{C_{down}} C_{\exp} tr(U \Sigma^{-1} U^\top U \Sigma U^\top) \\
=& \frac{C_{\exp}}{C_{down}} C_{\exp} tr(U U^\top) \\
=& \frac{C_{\exp}}{C_{down}} C_{\exp} tr(U^\top U) \\
=& \frac{C_{\exp}}{C_{down}} C_{\exp} tr(I_r) \\
=& \frac{C_{\exp}}{C_{down}} C_{\exp} r \\
\leq& \frac{C_{\exp}}{C_{down}} d .
\end{aligned}
\tag{30}
$$

Thus the Inequality 22 is proved. After that, we have:

$$\mathbb{E}_h[L(\tilde{\theta}_n)] - L(\theta^*)$$

$$= \mathbb{E}_h[L(\tilde{\theta}_n)] - \mathbb{E}_h[\hat{L}(\theta^*)]$$

$$\leq \mathbb{E}_h[L(\tilde{\theta}_n)] - \mathbb{E}_h[\hat{L}(\hat{\theta})]$$

$$= \frac{1}{n} \sum_{i=1}^{n} \mathbb{E}_h[L(\tilde{\theta}_i)] - \mathbb{E}_h[\hat{L}(\hat{\theta})]$$

$$= \frac{1}{n} \sum_{i=1}^{n} \mathbb{E}_h[l_i(\tilde{\theta}_i)] - \mathbb{E}_h[\hat{L}(\hat{\theta})]$$

$$= \frac{1}{n} \sum_{i=1}^{n} \mathbb{E}_h[l_i(\tilde{\theta}_i)] - \sum_{i=1}^{n} \mathbb{E}_h[l_i(\hat{\theta})]$$

$$= \frac{1}{n} \sum_{i=1}^{n} \mathbb{E}_h[l_i(\tilde{\theta}_i) - l_i(\hat{\theta})]$$

$$\leq \frac{C_{\exp}}{C_{down}} \frac{d}{n}$$

$$= O(\frac{d}{n})$$

Thus we has proved that $\mathbb{E}_h[L(\tilde{\theta}_n)] - L(\theta^*) \leq \frac{C_{\exp}}{C_{down}} \cdot \frac{d}{n}$. Notice that $\tilde{\theta}_n$ is generated by optimizing the leave-one-out likelihood function $\tilde{L}_n(\theta) = \sum_{j=1}^{n-1} l_j(\theta)$, which does not contain $l_n(\theta)$, and that the expected likelihood function $L(\theta)$ does not depend on any specific result occurring in this round. That is to say, every term of this inequality is not related to the last round $(x_n, v_n, \mathbb{1}_n)$ at all. In other words, this inequality is still valid if we only conduct this epoch from round 1 to $(n-1)$.

Now let $n = \tau + 1$, and then we know that $\tilde{\theta}_{\tau+1} = \hat{\theta}$. Therefore, the theorem holds. ∎

### B.3 Proof of Lower bound in Section 5.3

*Proof.* We assume a fixed $u^*$ such that $x^\top \theta^* = u^*, \forall x \in \mathbb{D}$. In other words, we are considering a non-context setting. Therefore, we can define a policy as $\Psi : \{0,1\}^t \to \mathbb{R}^+, t = 1, 2, \ldots$ that does not observe $x_t$ at all. Before the proof begins, we firstly define a few notations: We denote $\Phi_\sigma(\omega)$ and $p_\sigma(\omega)$ as the CDF and PDF of Gaussian distribution $\mathcal{N}(0, \sigma^2)$, and the corresponding $J_\sigma(u) = \arg\max_v v(1 - \Phi_\sigma(v - u))$ as the pricing function.

Since we have proved that $J'(u) \in (0, 1)$ for $u \in \mathbb{R}$ in Appendix B.2.2, we have the following lemma:

**Lemma 14.** $u - J_\sigma(u)$ *monotonically increases as* $u \in (0, +\infty), \forall \sigma > 0$. *Also, we know that* $J_\sigma(0) > 0, \forall \sigma > 0$.

Now consider the following cases: $\sigma_1 = 1, \sigma_2 = 1 - f(T)$, where $\lim_{T \to \infty} f(T) = 0, f'(T) < 0, 0 < f(T) < \frac{1}{2}$. We will later determine the explicit form of $f(T)$.

Suppose $u^*$ satisfies $J_{\sigma_1}(u^*) = u^*$. Solve it and get $u^* = \sqrt{\frac{\pi}{2}}$. Therefore, we have $u \in (0, u^*) \Leftrightarrow J_1(u) > u$, and $u \in (u^*, +\infty) \Leftrightarrow J_1(u) < u$. As a result, we have the following lemma.

**Lemma 15.** *For any* $\sigma \in (\frac{1}{2}, 1)$, *we have:*

$$J_\sigma(u^*) \in (0, u^*) \tag{31}$$

*Proof.* Firstly, we have:

$$J_\sigma(u) = \arg\max_v v\Phi_\sigma(u-v)$$

$$= \arg\max_v v\Phi_1(\frac{u-v}{\sigma})$$

$$= \arg\max_{\omega=\frac{v}{\sigma}} \sigma\omega\Phi_1(\frac{u}{\sigma}-\omega)$$

$$= \sigma \arg\max_\omega \Phi_1(\frac{u}{\sigma}-\omega)$$

$$= \sigma J_1(\frac{u}{\sigma}).$$

When $\sigma \in (\frac{1}{2}, 1)$, we know $\frac{u^*}{\sigma} > u^*$. Since $J_1(u^*) = u^*$ and that $u \in (u^*, +\infty) \Leftrightarrow J_1(u) < u$, we have $\frac{u^*}{\sigma} > J_1(\frac{u^*}{\sigma})$. Hence

$$u^* > \sigma J_1(\frac{u^*}{\sigma}) = J_\sigma(u^*). \tag{32}$$

∎

Therefore, without losing generality, we assume that for the problem parameterized by $\sigma_2$, the price $v \in (0, u^*)$. To be specific, suppose $v^*(\sigma) = J_\sigma(u^*)$. Define $\Psi_{t+1} : [0,1]^t \to (0, u^*)$ as any policy that proposes a price at time $t+1$. Define $\Psi = \{\Psi_1, \Psi_2, \ldots, \Psi_{T-1}, \Psi_T\}$.

Define the sequence of price as $V = \{v_1, v_2, \ldots, v_{T-1}, v_T\}$, and the sequence of decisions as $\mathbb{1} = \{\mathbb{1}_1, \mathbb{1}_2, \ldots, \mathbb{1}_{T-1}, \mathbb{1}_T\}$. Denote $V^t = \{v_1, v_2, \ldots, v_t, \}$.

Define the probability (also the likelihood if we change $u^*$ to other parameter $u$):

$$Q_T^{V,\sigma}(\mathbb{1}) = \prod_{t=1}^{T} \Phi_\sigma(u^* - v_t)^{\mathbb{1}_t} \Phi_\sigma(v_t - u^*)^{1-\mathbb{1}_t}. \tag{33}$$

Define a random variable $Y_t \in \{0,1\}^t$, $Y_t \sim Q_t^{V^t,\sigma}$ and one possible assignment $y_t = \{\mathbb{1}_1, \mathbb{1}_2, \ldots, \mathbb{1}_{t-1}, \mathbb{1}_t\}$. For any price $v$ and any parameter $\sigma$, define the expected reward function as $r(v, \sigma) := v\Phi_\sigma(u^* - v)$. Based on this, we can further define the expected regret $\text{Regret}(\sigma, T, \Psi)$:

$$\text{Regret}(\sigma, T, \Psi) = \mathbb{E}[\sum_{t=1}^{T} r(J_\sigma(u^*), \sigma) - r(\Psi_t(y_{t-1}), \sigma)] \tag{34}$$

Now we have the following properties:

**Lemma 16.**    *1.* $r(v^*(\sigma), \sigma) - r(v, \sigma) \geq \frac{1}{60}(v^*(\sigma) - v)^2$;

2. $|v^*(\sigma) - u^*| \geq \frac{2}{5}|1 - \sigma|$;

3. $|\Phi_\sigma(u^* - v) - \Phi_1(u^* - v)| \leq |u^* - v| \cdot |\sigma - 1|$.

*Proof.*    1. We have:

$$\frac{\partial r(v, \sigma)}{\partial v}\Big|_{v=v^*(\sigma)} = 0 \quad \frac{\partial^2 r(v, \sigma)}{\partial v^2} = \frac{1}{\sigma^2}(v^2 - u^*v - 2\sigma^2)p_\sigma(u^* - v)$$

Since $v \in (0, u^*)$, we have $(v^2 - u^*v - 2\sigma^2) < -2\sigma^2$. Also, since $\sigma \in (1/2, 1)$, we have $p_\sigma(u^* - v) > \frac{1}{\sqrt{2\pi}} \cdot e^{-\frac{(u^*)^2}{2\cdot(1/2)^2}} = \frac{1}{\sqrt{2\pi e^\pi}} > 0.017$. Therefore, we have

$$\frac{\partial^2 r(v, \sigma)}{\partial v^2} < -2 * 0.017 < -\frac{1}{30}$$

As a result, we have:

$$
\begin{aligned}
r(v*(\sigma),\sigma) - r(v,\sigma) &= -(v*(\sigma)-v)\frac{\partial r(v,\sigma)}{\partial v}\Big|_{v=v^*(\sigma)} - \frac{1}{2}(v*(\sigma)-v)^2\frac{\partial^2 r(v,\sigma)}{\partial v^2}\Big|_{v=\tilde{v}} \\
&= 0 - \frac{1}{2}(v*(\sigma)-v)^2\frac{\partial^2 r(v,\sigma)}{\partial v^2}\Big|_{v=\tilde{v}} \\
&\geq \frac{1}{2}\cdot\frac{1}{30}(v*(\sigma)-v)^2.
\end{aligned}
\tag{35}
$$

2. According to Equation 32, we know that:

$$
v^*(\sigma) = \sigma J_1(\frac{u^*}{\sigma})
$$

For $u \in (u^*, +\infty)$, $J_1(u) < u$. According to Lemma 14, we have:

$$
\begin{aligned}
J_1'(u) &= 1 + \frac{1}{J_1(u)(J_1(u)-u)-2} \\
&> 1 + \frac{1}{0-2} \\
&= \frac{1}{2}.
\end{aligned}
$$

Also, for $u \in (u^*, \frac{u^*}{\sigma})$, we have:

$$
\begin{aligned}
J_1'(u) &= 1 - \frac{1}{2+J_1(u)(u-J_1(u))} \\
&\underset{\substack{\uparrow \\ 0<J_1(u)<u}}{\leq} 1 - \frac{1}{2+u(u-0)} \\
&\underset{\substack{\uparrow \\ u<\frac{u^*}{\sigma}<\frac{u^*}{2}}}{\leq} 1 - \frac{1}{2+(\frac{u^*}{2})^2} \\
&= 1 - \frac{1}{2+\frac{\pi}{8}} \\
&< \frac{3}{5}.
\end{aligned}
$$

Therefore, we have:

$$
\begin{aligned}
J_1(u^*) - J_\sigma(u^*) &= J_1(u^*) - \sigma J_1(\frac{u^*}{\sigma}) \\
&= J_1(u^*) - \sigma J_1(u^*) + \sigma(J_1(u^*) - J_1(\frac{u^*}{\sigma})) \\
&= J_1(u^*)(1-\sigma) - \sigma(J_1(\frac{u^*}{\sigma}) - J_1(u^*)) \\
&> u^*(1-\sigma) - \sigma\cdot\frac{3}{5}(\frac{u^*}{\sigma}-u^*) \\
&= u^*(1-\sigma) - \frac{3}{5}\sigma(\frac{1}{\sigma}-1)u^* \\
&= u^*(1-\sigma)(1-\frac{3}{5}) \\
&> \frac{2}{5}(1-\sigma).
\end{aligned}
$$

3. This is because:
$$\begin{aligned}
|\Phi_\sigma(u^* - v) - \Phi_1(u^* - v)| &= |\Phi_\sigma(u^* - v) - \Phi_\sigma(\sigma(u^* - v))| \\
&\leq \max |p_\sigma| \cdot |(u^* - v) - \sigma(u^* - v)| \\
&\leq \frac{1}{\sqrt{2\pi}\sigma} \cdot |(u^* - v) - \sigma(u^* - v)| \\
&\underset{\underset{\sigma > \frac{1}{2}}{\uparrow}}{\leq} (1 - \sigma)|u^* - v|.
\end{aligned}$$

∎

In the following part, we will propose two theorems, which balance the cost of learning and that of uncertainty. This part is mostly similar to [BR12] Section 3, but we adopt a different family of demand curves here.

**Theorem 17** (Learning is costly). *Let $\sigma \in (1/2, 1)$ and $v_t \in (0, u^*)$, and we have:*
$$\mathcal{K}(Q^{V,1}; Q^{V,\sigma}) < 9900(1 - \sigma)^2 \text{Regret}(1, T, \Psi). \tag{36}$$
*Here $v_t = \Psi(y_{t-1}), t = 1, 2, \dots, T$.*

*Proof.* First of all, we cite the following lemma that would facilitate the proof.

**Lemma 18** (Corollary 3.1 in Taneja and Kumar, 2004). *Suppose $B_1$ and $B_2$ are distributions of Bernoulli random variables with parameters $q_1$ and $q_2$, respectively, with $q_1, q_2 \in (0, 1)$. Then,*
$$\mathcal{K}(B_1; B_2) \leq \frac{(q_1 - q_2)^2}{q_2(1 - q_2)}.$$

According to the definition of KL-divergence, we have:
$$\mathcal{K}(Q_T^{V,1}; Q_T^{V,\sigma}) = \sum_{s=1}^{T} \mathcal{K}(Q_s^{V^s,1}; Q_s^{V^s,\sigma}|Y_{s-1}).$$

For each term of the RHS, we have:
$$\begin{aligned}
&\mathcal{K}(Q_s^{V^s,1}, Q_s^{V^s,\sigma}|Y_{s-1}) \\
&= \sum_{y_s \in \{0,1\}^s} Q_s^{V^s,1}(y_s) \log\left(\frac{Q_s^{V^s,1}(\mathbb{1}_s|y_{s-1})}{Q_s^{V^s,\sigma}(\mathbb{1}_s|y_{s-1})}\right) \\
&\underset{\underset{\text{split } y_s \text{ as } y_{s-1} \text{ and } ind_s}{\uparrow}}{=} \sum_{y_{s-1} \in \{0,1\}^{s-1}} Q_{s-1}^{V^{s-1},1}(y_{s-1}) \cdot \sum_{\mathbb{1}_s \in \{0,1\}} Q_s^{V^s,1}(\mathbb{1}_s|y_{s-1}) \log\left(\frac{Q_s^{V^s,1}(\mathbb{1}_s|y_{s-1})}{Q_s^{V^s,\sigma}(\mathbb{1}_s|y_{s-1})}\right) \\
&= \sum_{y_{s-1} \in \{0,1\}^{s-1}} Q_{s-1}^{V^{s-1},1}(y_{s-1})\mathcal{K}\left(Q_s^{V^s,1}(\cdot|y_{s-1}), Q_s^{V^s,\sigma}(\cdot|y_{s-1})\right) \\
&\underset{\underset{\text{Lemma } 18}{\uparrow}}{\leq} \sum_{y_{s-1} \in \{0,1\}^{s-1}} Q_{s-1}^{V^{s-1},1}(y_{s-1})\frac{(\Phi_1(u^* - v_s) - \Phi_\sigma(u^* - v_s))^2}{\Phi_\sigma(u^* - v_s)(1 - \Phi_\sigma(u^* - v_s))} \\
&= \frac{1}{\Phi_\sigma(u^* - v_s)(1 - \Phi_\sigma(u^* - v_s))} \sum_{y_{s-1} \in \{0,1\}^{s-1}} Q_{s-1}^{V^{s-1},1}(y_{s-1})(\Phi_1(u^* - v_s) - \Phi_\sigma(u^* - v_s))^2 \\
&\underset{\underset{(**)}{\uparrow}}{\leq} 165 \cdot \sum_{y_{s-1} \in \{0,1\}^{s-1}} Q_{s-1}^{V^{s-1},1}(y_{s-1})(\Phi_1(u^* - v_s) - \Phi_\sigma(u^* - v_s))^2 \\
&\underset{\underset{\text{Lemma } 16 \text{ Property } 3}{\uparrow}}{\leq} 165 \cdot \sum_{y_{s-1} \in \{0,1\}^{s-1}} Q_{s-1}^{V^{s-1},1}(y_{s-1})(u^* - v_s)^2(1 - \sigma)^2 \\
&= 165(1 - \sigma)^2 \mathbb{E}_{Y_{s-1}}[(u^* - v_s)^2].
\end{aligned}$$

Here inequality (**) above is proved as follows: since $v_s \in (0, u^*)$ as is assumed, we have:

$$\frac{1}{2} < \Phi_\sigma(u^* - v_s) < \Phi_\sigma(u^*)$$

$$= \sigma \cdot \Phi_1(\frac{u^*}{\sigma})$$

$$\leq 1 \cdot \Phi_1(\frac{\sqrt{\frac{\pi}{2}}}{\frac{1}{2}})$$

$$\leq \Phi_1(\sqrt{2\pi})$$

$$\leq 0.9939 .$$

As a result, we have $\frac{1}{\Phi_\sigma(u^* - v_s)(1 - \Phi_\sigma(u^* - v_s))} \leq \frac{1}{0.9939 \times 0.0061} = 164.7988 \leq 165$. Therefore, by summing up all $s$, we have:

$$\mathcal{K}(Q_T^{V,1}; Q_T^{V,\sigma}) = \sum_{s=1}^{T} \mathcal{K}(Q_s^{V^s,1}; Q_s^{V^s,\sigma}|Y_{s-1})$$

$$\leq 165(1 - \sigma)^2 \sum_{s=1}^{T} \mathbb{E}_{Y_{s-1}}[(u^* - v_s)^2]$$

$$\underset{\substack{\uparrow \\ \text{Lemma 16 Property 1}}}{\leq} 165 \times 60 \cdot (1 - \sigma)^2 \sum_{s=1}^{T} (r(u^*, 1) - r(v_s, 1))$$

$$\underset{\substack{\uparrow \\ \text{definition of regret and } v_s = \Psi(y_{s-1}).}}{=} 9900(1 - \sigma)^2 \text{Regret}(1, T, \Psi),$$

which concludes the proof. ∎

**Theorem 19** (Uncertainty is costly). *Let $\sigma \leq 1 - T^{-\frac{1}{4}}$, and we have:*

$$\text{Regret}(1, T, \Psi) + \text{Regret}(\sigma, T, \Psi) \geq \frac{1}{24000} \cdot \sqrt{T} \cdot e^{-\mathcal{K}(Q^{V,1}; Q^{V,\sigma})}. \tag{37}$$

*Here $v_t = \Psi(y_{t-1}), t = 1, 2, \ldots, T$.*

*Proof.* First of all, we cite a lemma that would facilitate our proof:

**Lemma 20.** *Let $Q_0$ and $Q_1$ be two probability distributions on a finite space $\mathcal{Y}$; with $Q_0(y), Q_1(y) > 0, \forall y \in \mathcal{Y}$. Then for any function $F : \mathcal{Y} \to \{0, 1\}$,*

$$Q_0\{F = 1\} + Q_1\{J = 0\} \geq \frac{1}{2} e^{-\mathcal{K}(Q_0; Q_1)},$$

*where $\mathcal{K}(Q_0; Q_1)$ denotes the KL-divergence of $Q_0$ and $Q_1$.*

Define two intervals of prices:

$$C_1 = \{v : |u^*| \leq \frac{1}{10T^{\frac{1}{4}}}\} \ \text{and} \ C_2 = \{v : |J_\sigma(u^*) - v| \leq \frac{1}{10T^{\frac{1}{4}}}\}$$

Note that $C_1$ and $C_2$ are disjoint, since $|u^* - J_\sigma(u^*)| \geq \frac{2}{5}|1 - \sigma| = \frac{2}{5T^{1/2}}$ according to Lemma 16 Property 2. Also, for $v \in (0, u^*)\backslash C_2$, the regret is large according to Lemma 16 Property 1, because:

$$r(v^*(\sigma), \sigma) - r(v, \sigma) \geq \frac{1}{60}(v - v^*(\sigma))^2 \geq \frac{1}{6000T^{\frac{1}{2}}}.$$

Then, we have:

$$\text{Regret}(1, T, \Psi) + \text{Regret}(\sigma, T, \Psi)$$

$$\geq \sum_{t=1}^{T-1} \mathbb{E}_1[r(u^*, 1) - r(v_{t+1}, 1)] + \mathbb{E}_\sigma[r(J_\sigma(u^*), \sigma) - r(v_{t+1}, \sigma)]$$

$$\geq \frac{1}{6000\sqrt{T}} \sum_{t=1}^{T-1} \mathbb{P}_1[v_{t+1} \notin C_1] + \mathbb{P}_\sigma[v_{t+1} \notin \{C_2\}]$$

$$\underset{\underset{Suppose \ F_{t+1}=\mathbb{1}[v_{t+1}\in C_2]}{\uparrow}}{\geq} \frac{1}{6000\sqrt{T}} \sum_{t=1}^{T-1} \mathbb{P}_1[F_{t+1} = 1] + \mathbb{P}_\sigma[F_{t+1} = 0]$$

$$\underset{\underset{Lemma \ 20}{\uparrow}}{\geq} \frac{1}{6000\sqrt{T}} \sum_{t=1}^{T-1} \frac{1}{2} e^{-\mathcal{K}(Q_t^{V^t,1}; Q_t^{V^t,\sigma})}$$

$$\underset{\underset{\mathcal{K}(Q_t^{V^t,1}; Q_t^{V^t,\sigma}) \ not \ decreasing}{\uparrow}}{\geq} \frac{1}{6000\sqrt{T}} \frac{T-1}{2} e^{-\mathcal{K}(Q_T^{V,1}; Q_T^{V,\sigma})}$$

$$\geq \frac{1}{24000} \sqrt{T} e^{-\mathcal{K}(Q_T^{V,1}; Q_T^{V,\sigma})}.$$

∎

According to Theorem 17 and Theorem 19, we can then prove Theorem 12. Let $\sigma = 1 - T^{-\frac{1}{4}}$

$$2 \left(\text{Regret}(1, T, \Psi) + \text{Regret}(\sigma, T, \Psi)\right)$$

$$\geq \text{Regret}(1, T, \Psi) + (\text{Regret}(1, T, \Psi) + \text{Regret}(\sigma, T, \Psi))$$

$$\geq \frac{1}{9900T^{-1/2}} \mathcal{K}(Q^{V,1}; Q^{V,\sigma}) + \frac{1}{24000} \cdot \sqrt{T} \cdot e^{-\mathcal{K}(Q^{V,1}; Q^{V,\sigma})}$$

$$\geq \frac{1}{24000} \sqrt{T} \left( \mathcal{K}(Q^{V,1}; Q^{V,\sigma}) + e^{-\mathcal{K}(Q^{V,1}; Q^{V,\sigma})} \right)$$

$$\underset{\underset{The \ fact \ e^x \geq x+1, \forall x \in \mathbb{R}}{\uparrow}}{\geq} \frac{1}{24000} \sqrt{T}.$$

Thus Theorem 12 is proved valid. ∎

## C  More Discussions

### C.1  Dependence on $B$ and Noise Variance

Here we use a concrete example to analyze the coefficients of regret bounds. Again, we assume that $N_t \sim \mathcal{N}(0, \sigma^2)$. Notice that both $C_s$ and $C_a$ have a component of $\frac{C_{\exp}}{C_{down}}$. In order to analyze $\frac{C_{\exp}}{C_{down}}$, we define a *hazard function* denoted as $\lambda(\omega)$ with $\omega \in \mathbb{R}$:

$$\lambda(\omega) := \frac{p_1(\omega)}{1 - \Phi_1(\omega)} = \frac{p_1(-\omega)}{\Phi_1(-\omega)}, \tag{38}$$

where $\Phi_1$ and $p_1$ are the CDF and PDF of standard Gaussian distribution. The concept of hazard function comes from the area of *survival analysis*. From Equation 11 and 13, we plug in Equation 38 and get:

$$C_{\text{down}} \geq \inf_{\omega \in [-\frac{B}{\sigma}, \frac{B}{\sigma}]} \left\{ \frac{1}{\sigma^2} \lambda(-\omega)^2 + \omega \cdot \lambda(-\omega) \right\}$$

$$C_{\exp} \leq \sup_{\omega \in [-\frac{B}{\sigma}, \frac{B}{\sigma}]} \left\{ \frac{1}{\sigma^2} \lambda(-\omega)^2 \right\}. \tag{39}$$

In Lemma 21, we will prove that $\lambda(\omega)$ is exponentially small as $\omega \to +\infty$, and is asymptotically close to $-\omega$ as $\omega \to -\infty$. Therefore, $C_{down}$ is exponentially small and $C_{exp}$ is quadratically large with respect to $B/\sigma$. Although we assume that $B$ and $\sigma$ are constant, we should be alert that the scale of $B/\sigma$ can be very large as $\sigma$ goes to zero, i.e. as the noise is "insignificant". In practice (especially when $T$ is finite), this may cause extremely large regret at the beginning. A "Shallow Pricing" method introduced by Cohen et al. [2020] (as well as other domain-cutting methods in contextual searching) may serve as a good pre-process as it frequently conducts bisections to cut the feasible region of $\theta^*$ with high probability. According to Theorem 3 in Cohen et al. [2020], their Shallow Pricing algorithm will bisect the parameter set for at most logarithmic times to ensure that $\frac{B}{\sigma}$ has been small enough (i.e. upper-bounded by $O(poly \log(T))$). However, this does not necessarily means that we can use a $O(\log T)$-time pre-process to achieve the same effect, since they run the algorithm throughout the session while we only take it as a pre-process. Intuitively, at least under the adversarial feature assumption, we cannot totally rely on a few features occurring at the beginning (as they might be misleading) to cut the parameter set once and for all. A mixture approach of Shallow Pricing and EMLP/ONSP might work, as the algorithm can detect whether current $\frac{B}{\sigma}$ is larger than a threshold of bisection. However, this requires new regret analysis as the operations parameter domain are changing over time. Therefore, we claim in Section 7 that the regret bound is still open if $\sigma = \Theta(T^{-\alpha})$ for $\alpha \in (0, 1)$.

**Lemma 21** (Properties of $\lambda(\omega)$). *For* $\lambda(\omega) := \frac{p_1(\omega)}{1 - \Phi_1(\omega)}$*, we have:*

1, $\quad \dfrac{d}{d\omega}\lambda(\omega) > 0.$

2, $\quad \lim_{\omega \to -\infty} \omega^k \lambda(\omega) = 0, \ \forall k > 0.$

3, $\quad \lim_{\omega \to +\infty} \lambda(\omega) - \omega = 0.$

4, $\quad \lim_{\omega \to +\infty} \omega \left( \lambda(\omega) - \omega \right) = 1.$

*Proof.* We prove the Lemma 21 sequentially:

1. We have:

$$
\begin{aligned}
\lambda'(\omega) =& \frac{p_1^2(-\omega) - p_1'(-\omega)\Phi_1(-\omega)}{\Phi_1(-\omega)^2} \\
=& \frac{p_1^2(-\omega) - \omega p_1(-\omega)\Phi_1(-\omega)}{\Phi_1(-\omega)^2} \\
=& \frac{p_1(-\omega)\left(p_1(-\omega) - \omega\Phi_1(-\omega)\right)}{\Phi_1(-\omega)^2}.
\end{aligned}
\tag{40}
$$

Therefore, it is equivalent to prove that $p_1(-\omega) - \omega\Phi_1(-\omega) > 0$.

Suppose $f(\omega) = p_1(\omega) + \omega\Phi_1(\omega)$. We now take its derivatives as follows:

$$
\begin{aligned}
f'(\omega) &= p_1'(\omega) + (\Phi_1(\omega) + \omega \cdot p_1(\omega)) \\
&= (-\omega)p_1(\omega) + \Phi_1(\omega) + \omega \cdot p_1(\omega) \\
&= \Phi_1(\omega) \\
&> 0
\end{aligned}
\tag{41}
$$

Therefore, we know that $f(\omega)$ monotonically increases in $\mathbb{R}$. Additionally, since we have:

$$
\begin{aligned}
&\lim_{\omega\to-\infty} f(\omega) \\
&= \lim_{\omega\to-\infty} p_1(\omega) + \lim_{\omega\to-\infty} \omega\Phi_1(\omega) \\
&= 0 + \lim_{\omega\to-\infty} \frac{1}{\sigma^2}\cdot\frac{\Phi_1(\omega)}{1/\omega} \\
&= \lim_{\omega\to-\infty} \cdot\frac{p_1(\omega)}{-1/\omega^2} \\
&= \lim_{\omega\to-\infty} \cdot\left(-\frac{1}{\sqrt{2\pi}}\cdot\frac{\omega^2}{\exp\{\frac{\omega^2}{2}\}}\right) \\
&= 0
\end{aligned}
\tag{42}
$$

Therefore, we know that $f(\omega) > 0, \forall \omega \in \mathbb{R}$, and as a result, $\lambda'(\omega) > 0$.

2. We have:

$$
\begin{aligned}
&\lim_{\omega\to-\infty} \omega^k \lambda(\omega) \\
&= \lim_{\omega\to-\infty} \omega^k \frac{p_1(-\omega)}{\Phi_1(-\omega)} \\
&= \frac{\lim_{\omega\to-\infty} \omega^k p_1(-\omega)}{\lim_{\omega\to-\infty} \Phi_1(-\omega)} \\
&= \frac{\lim_{\omega\to-\infty} \omega^k(\frac{1}{\sqrt{2\pi}}\exp\{-\frac{\omega^2}{2}\})}{1} \\
&= \frac{0}{1} \\
&= 0.
\end{aligned}
\tag{43}
$$

3. We only need to prove that

$$
\lim_{\omega\to+\infty} \lambda(\omega) - \omega = 0.
$$

Actually, we have:

$$
\begin{aligned}
&\lim_{\omega\to+\infty} \lambda(\omega) - \omega \\
&= \lim_{\omega\to+\infty} \frac{p_1(-\omega) - \omega\Phi_1(-\omega)}{\Phi_1(-\omega)} \\
&= \lim_{\omega\to-\infty} \frac{p_1(\omega) + \omega\Phi_1(\omega)}{\Phi_1(\omega)} \\
&\overset{\text{L'Hospital's rule}}{=} \lim_{\omega\to-\infty} \frac{(-\omega)p_1(\omega) + \Phi_1(\omega) + \omega p_1(\omega)}{p_1(\omega)} \\
&= \lim_{\omega\to-\infty} \frac{\Phi_1(\omega)}{p_1(\omega)} \\
&= \lim_{\omega\to-\infty} \frac{p_1(\omega)}{(-\omega)p_1(\omega)} \\
&= 0
\end{aligned}
\tag{44}
$$

4.

$$
\begin{aligned}
&\lim_{\omega \to +\infty} \omega(\lambda(\omega) - \omega) \\
&= \lim_{\omega \to +\infty} \frac{\omega\left(p_1(-\omega) - \omega\Phi_1(-\omega)\right)}{\Phi_1(-\omega)} \\
&= \lim_{\omega \to -\infty} \frac{-\omega p_1(\omega) - \omega^2 \Phi_1(\omega)}{\Phi_1(\omega)} \\
&\overset{\text{L'Hospital's rule}}{\underset{\downarrow}{=}} \lim_{\omega \to -\infty} \frac{-p_1(\omega) - \omega(-\omega)p_1(\omega) - \omega^2 p_1(\omega) - 2\omega \cdot \Phi_1(\omega)}{p_1(\omega)} \\
&= -1 - 2\lim_{\omega \to -\infty} \frac{\omega\Phi_1(\omega)}{p_1(\omega)} \\
&= -1 + 2\lim_{\omega \to +\infty} \frac{1}{\frac{\lambda(\omega)}{\omega}} \\
&= -1 + 2 \\
&= 1.
\end{aligned} \tag{45}
$$

Thus the lemma holds.

∎

## C.2 Algorithmic Design

### C.2.1 Probit and Logistic Regressions

A probit/logit model is described as follows: a Boolean random variable $Y$ satisfies the following probabilistic distribution: $\mathbb{P}[Y = 1|X] = F(X^\top \beta)$, where $X \in \mathbb{R}$ is a random vector, $\beta \in \mathbb{R}$ is a parameter, and $F$ is the cumulative distribution function (CDF) of a (standard) Gaussian/logistic distribution. In our problem, we may treat $\mathbb{1}_t$ as $Y$, $[x_t^\top, v_t]^\top$ as $X$ and $[\theta^{*\top}, -1]^\top$ as $\beta$, which exactly fits this model if we assume the noise as Gaussian or logistic. Therefore, $\hat{\theta}_k = \arg\min_\theta \hat{L}_k(\theta)$ can be solved via the highly efficient implementation of generalized linear models, e.g., GLMnet, rather than resorting to generic tools for convex programming. As a heuristic, we could leverage the vast body of statistical work on probit or logit models and adopt a fully Bayesian approach that jointly estimates $\theta$ and hyper-parameters of $F$. This would make the algorithm more practical by eliminating the need to choose the hyper-parameters when running this algorithm.

### C.2.2 Advantages of EMLP over ONSP.

For the stochastic setting, we specifically propose EMLP even though ONSP also works. This is because EMLP only "switch" the pricing policy $\hat{\theta}$ for $\log T$ times. This makes it appealing in many applications (especially for brick-and-mortar sales) where the number of policy updates is a bottleneck. In fact, the iterations within one epoch can be carried out entirely in parallel.

### C.2.3 Agnostic Dynamic Pricing: Explorations versus Exploitation

At the moment, the proposed algorithm relies on the assumption of a linear valuation function (see Appendix C.3 for more discussion on problem modeling). It will be interesting to investigate the settings of model-misspecified cases and the full agnostic settings. The key would be to exploit the structural feedback in model-free policy-evaluation methods such as importance sampling. The main reason why we do not explore lies in the noisy model: essentially we are implicitly exploring a higher (permitted) price using the naturally occurring noise in the data. In comparison, there is another problem setting named "adversarial irrationality" where some of the customers will valuate the product adaptively and adversarially[10]. Existing work Krishnamurthy et al. [2021] adopts this setting and shows a linear regret dependence on the number of irrational customers, but they consider a different loss function (See Related Works Section).

---

[10]An adaptive adversary may take actions adversarially in respond to the environmental changes. In comparison, what we allow for the "adversarial features" is actually chosen by an oblivious adversary before the interactions start.

### C.3 Problem Modeling

#### C.3.1 Noise Distributions

In this work, we have made four assumptions on the noise distribution: strict log-concavity, $2^{nd}-$ order smooth, known, and i.i.d.. Here we explain each of them specifically.

- The assumption of knowing the exact $F$ is critical to the regret bound: If we have this knowledge, then we achieve $O(\log T)$ even with adversarial features; otherwise, an $\Omega(\sqrt{T})$ regret is unavoidable even with stochastic features.

- The strictly log-concave distribution family includes Gaussian and logistic distributions as two common noises. In comparison, Javanmard and Nazerzadeh [2019] assumes log-concavity that further covers Laplacian, exponential and uniform distributions. Javanmard and Nazerzadeh [2019] also considers the cases when (1) the noise distribution is unknown but log-concave, and (2) the noise distribution is zero-mean and bounded by support of $[-\delta, \delta]$. For case (1), they propose an algorithm with regret $O(\sqrt{T})$ and meanwhile prove the same lower bound. For case (2), they propose an algorithm with linear regret.

- The assumption that $F$ is $2^{nd}-$order smooth is also assumed by Javanmard and Nazerzadeh [2019] by taking derivatives $f'(v)$ and applying its upper bound in the proof. Therefore, we are still unaware of the regret bound if the noise distribution is discrete, where a lower bound of $\Omega(\sqrt{T})$ can be directly applied from Kleinberg and Leighton [2003].

- We even assume that the noise is identically distributed. However, the noise would vary among different people. The same problem happens on the parameter $\theta^*$: can we assume different people sharing the same evaluation parameter? We may interpret it in the following two ways, but there are still flaws: (1) the "customer" can be the public, i.e. their performance is quite stable in general; or (2) the customer can be the same one over the whole time series. However, the former explanation cannot match the assumption that we just sell one product at each time, and the latter one would definitely undermine the independent assumption of the noise: people would do "human learning" and might gradually reduce their noise of making decisions. To this extent, it is closer to the fact if we assume noises as martingales. This assumption has been stated in Qiang and Bayati [2016].

#### C.3.2 Linear Valuations on Features

There exist many products whose prices are not linearly dependent on features. One famous instance is a diamond: a kilogram of diamond powder is very cheap because it can be produced artificially, but a single 5-carat (or 1 gram) diamond might cost more than \$100,000. This is because of an intrinsic non-linear property of diamond: large ones are rare and cannot be (at least easily) compound from smaller ones. Another example lies in electricity pricing [Joskow and Wolfram, 2012], where the more you consume, the higher unit price you suffer. On the contrary, commodities tend to be cheaper than retail prices. These are both consequences of marginal costs: a large volume consuming of electricity may cause extra maintenance and increase the cost, and a large amount of purchasing would release the storage and thus reduce their costs. In a word, our problem setting might not be suitable for those large-enough features, and thus an upper bound of $x^\top \theta$ becomes a necessity.

### C.4 *Ex Ante* v.s. *Ex Post* Regrets

In this work, we considered the *ex ante* regret $Reg_{ea} = \sum_{t=1}^T \max_\theta \mathbb{E}[v_t^\theta \cdot \mathbb{1}(v_t^\theta \le w_t)] - \mathbb{E}[v_t \cdot \mathbb{1}(v_t \le w_t)]$, where $v_t^\theta = J(x_t^\top \theta)$ is the greedy price with parameter $\theta$ and $w_t = x_t^\top \theta^* + N_t$ is the realized random valuation. The *ex post* definition of the cumulative regret, i.e., $Reg_{ep} = \max_\theta \sum_{t=1}^T v_t^\theta \mathbb{1}(v_t^\theta \le w_t) - v_t \mathbb{1}(v_t \le w_t)$ makes sense, too. Note that we can decompose $\mathbb{E}[Reg_{ep}] = Reg_{ea} + \mathbb{E}[\max_\theta \sum_{t=1}^T v_t^\theta \mathbb{1}(v_t^\theta \le w_t) - \sum_{t=1}^T v_t^{\theta^*} \mathbb{1}(v_t^{\theta^*} \le w_t)]$. While it might be the case that the second term is $\Omega(\sqrt{dT})$ as the reviewer pointed out, it is a constant independent of the algorithm. For this reason, we believe using $Reg_{ea}$ is without loss of generality, and it reveals more nuanced performance differences of different algorithms.

For an *ex post dynamic* regret, i.e., $Reg_d = \sum_{t=1}^T w_t - v_t \cdot \mathbb{1}(v_t \le w_t)$, it is argued in Cohen et al. [2020] that any policy must suffer an expected regret of $\Omega(T)$ (even if $\theta^*$ is known). We may also present a good example lies in $N_t \sim \mathcal{N}(0,1), x_t^\top \theta^* = \sqrt{\frac{\pi}{2}}$ where the optimal price is $\sqrt{\frac{\pi}{2}}$

as well but the probability of acceptance is only 1/2, and this leads to a constant *per-step* regret of $\frac{1}{2}\sqrt{\frac{\pi}{2}}$.

### C.5  Ethic Issues

A field of study lies in "personalized dynamic pricing" [Aydin and Ziya, 2009, Chen and Gallego, 2021], where a firm makes use of information of individual customers and sets a unique price for each of them. This has been frequently applied in airline pricing [Krämer et al., 2018]. However, this causes first-order pricing discrimination. Even though this "discrimination" is not necessarily immoral, it must be embarrassing if we are witted proposing the same product with different prices towards different customers. For example, if we know the coming customer is rich enough and is not as sensitive towards a price (e.g., he/she has a variance larger than other customers), then we are probably raising the price without being too risky. Or if the customer is used to purchase goods from ours, then he or she might have a higher expectation on our products (e.g., he/she has a $\theta = a\theta^*, a > 1$), and we might take advantage and propose a higher price than others. These cases would not happen in an auction-based situation (such as a live sale), but might frequently happen in a more secret place, for instance, a customized travel plan.