# OpenReview forum: "Logarithmic Regret in Feature-based Dynamic Pricing"
_NeurIPS.cc/2021/Conference — NeurIPS 2021 Spotlight_

### Official Review · Reviewer_FNhv · 2021-07-13

**Rating:** 9
**Confidence:** 4

**Summary:**

This paper studies the problem of feature-based dynamic pricing and offers strong new regret bounds. The headline result is the construction of Online Newton Step Pricing and the proof that it obtains regret $O(d \log T)$ even with adversarial features. This had been proven before only by assuming either (1) small noise or (2) iid features plus a minimum-eigenvalue assumption.

**Limitations And Societal Impact:**

As for limitations, the paper  makes an intriguing point in Section 7: the regret bound of ONSP and EMLP becomes worse as the noise becomes smaller. Noise is helpful because it makes the regret smoother and induces exploration. Therefore, their analysis does not apply if the noise is $O(T^{-\alpha})$ for $\alpha > 0$. Interestingly, we now have logarithmic regret bounds for $\alpha = 0$ from this paper and $\alpha = 1$ from Cohen, Lobel and Paes Leme, but we do not have logarithmic regret bounds for $\alpha \in (0,1)$.

As for potential negative societal impacts, the same algorithms that are proposed here for pricing a large collection of products could alternatively be used for personalized pricing, which could potentially be problematic depending on the situation. This point is properly disclosed in the appendix.

**Main Review:**


This paper studies the problem of feature-based (or contextual) dynamic pricing. There exists an unknown vector $\theta$ that characterizes a consumer's preferences. At every period $t=1,...,T$, a new product is on offer with a set of characteristics described by $x_t$. The consumer's valuation for this product is given by $\theta^\top x_t + N_t$, where $N_t$ is some iid noise from a known distribution. The problem is how to set a price $v_t$ such that the product will be sold if and only if $v_t \leq \theta^\top x_t + N_t$.

There are two different variations of this problem depending on whether the feature vectors are drawn iid or are generated adversarially. For the adversarial features problem, Cohen, Lobel and Paes Leme (EC 2016 and MS 2020) proved a $O(poly(\log T))$ regret bound assuming the noise $N_t$ is small (of order $O(1/(T\log T))$. For the noise-free case, Paes Leme and Schneider (FOCS 2018) proved a $O(\log \log T)$ regret bound. For the iid case, Javanmard and Nazerzadeh (JMLR 2019) proved a $O(\log T)$ bound assuming the covariance matrix of the feature vectors has a smallest eigenvalue bounded away from zero. Note that, following the template of the discussion in the paper being evaluated, I am ignoring the dependance on the feature dimension $d$ in this paragraph.

This paper's main contribution is an algorithm called Online Newton Step Pricing (ONSP) that obtains $O(d\log T)$ regret for the adversarial version of this problem, without the small noise assumption (noise is assumed to be $O(1)$). The idea is pretty simple (though the proof isn't): use the log-likelihood as a loss function and then use online Newton's method to optimize it.

The paper also presents a separate algorithm for the iid case, called the Epoch-based Max-Likelihood Pricing (EMLP) algorithm. This algorithm is the exact same as the one proposed by Javanmard and Nazerzadeh, except it replaces a Lasso regression with a standard one. They prove the same $O(d\log T)$ regret as Javanmard and Nazerzadeh for the iid case using this algorithm, without invoking any sort of smallest-eigenvalue assumption. The paper further proves that if the noise distribution is unknown, no bound stronger than $O(\sqrt T)$ is attainable.

This paper is impressive, and I support its publication at NeurIPS. I would say the paper is original, high-quality, clear and significant. Here is some feedback and questions:

1. What is the point of EMLP? ONSP can obviously also be used with iid features. Does EMLP offer any advantages?

2. The paper clearly dominates the earlier results of Javanmard and Nazerzadeh. They obtain the same regret bound without the eigenvalue assumption. They even do so with the same algorithm (EMLP). The main exception to the point is that the current paper does not have sparsity-related results, as it dropped the Lasso regression.

3. The paper also solves an open question in this literature: is it possible to obtain $O(d\log T)$ regret with $O(1)$ noise? They paper settles this issues affirmatively. In my opinion, this is an important result.

Typo: "vice versa" not "vise versa" on page 3.


**Time Spent Reviewing:**

8

---

> ### Author Response · Authors · 2021-08-06
> **Response to Reviewer FNhv**
>
> Thank you for your encouraging comments! We will incorporate your comments and give multiple editing passes to catch and correct all typos.
>
> 1, It is true that ONSP is suitable for all settings in which EMLP does well. However, EMLP only has to update the pricing policy (i.e. the parameter $\hat{\theta}_t$) for logarithmic times, i.e., it has a low switching-cost. In contrast, ONSP requires updating the policy in every iteration. As we discussed in Appendix C.2.2., updating the pricing policy could be a bottleneck in brick-and-mortar sales, and EMLP is thus a more practical algorithm in stochastic environments.
>
> 2, We did not consider the sparsity in this paper, since we mainly focus on the relatively low-dimensional settings and compare with existing results with respect to dependence on $T$.  A generalization to the sparse / high-dimensional setting is potentially tractable using our techniques --- and it would be quite an interesting result in our opinion if a distribution-free logarithmic regret can be derived ---  but is beyond the scope of this paper.

---

### Official Review · Reviewer_XCzw · 2021-07-14

**Rating:** 6
**Confidence:** 3

**Summary:**

This paper considers the feature-based dynamic pricing setting. This setting is a sequential setting in which in each round a learner is to predict the price of a product based on a feature vector. Then, if the customer's private value exceeds the price predicted by the learner, the customer buys the product. The goal is to control the difference between the difference between the cumulative price of the sold products and the best offline pricing policy.

On of the main challenge in this setting lies in the feedback that the learner receives. Since the learner only sees whether or not a product is sold, he receives limited feedback about the customers private value for a product. To remediate this issue, the authors assume that the private value of the customer is the inner product between a unknown parameter vector and the feature vector plus some noise. Another important assumption is that the distribution of the the noise is known exactly and that both the cumulative distribution $F$ and $1-F$ are strictly log-concave.

With this set of assumptions (plus some additional standard assumptions) the authors provide two new algorithms, one for stochastic feature vectors and one for adversarial feature vectors. Both algorithms rely on the fact that as soon as both $F$ and $1-F$ are strictly log-concave and $F$ is known, one can derive a surrogate loss which is almost quadratic. Because of the nice properties of this surrogate loss functions to show that the regret is $O(d \ln(T))$.

The paper also contains a lower bound, roughly stating that as soon as the distribution of the noise is not exactly known, the regret has a $O(\sqrt{T})$ lower bound.

Finally, the paper contains some experiments which compare the performance of the two new algorithms to EXP-4.

**Limitations And Societal Impact:**

The authors adequately addressed the limitations and potential negative societal impact of their work.

**Main Review:**

The idea of using the log-concavity of the noise is nice and turns out to be quite effective in deriving small regret bounds.

One of the arguments for using the new methods over previous methods is that existing results have regret bounds that scale are $O(\min\{\frac{1}{\lambda^2_{min}}\ln(T), \sqrt{T}\})$, where $\lambda_{min}$ is the smallest eigenvalue of the second moment matrix of the feature vectors, which can be arbitrarily small depending on the distribution of the feature vectors. Since there are also constants that depend on a distribution in the new bounds it would be nice to have some idea of how big this constants are. While in appendix C there are some lower bounds for the constants I would like to seem an upper bound as well to get an idea of how the new regret bounds compare with the literature. Also, note that for the Newton algorithm to work one needs to know the exp-concavity parameter of the surrogate losses, which seem to be tied to the same constants. So for the algorithm to have any practical relevance it would be a good idea to provide the constants for some common distributions.

On a similar note, I think the statement in line 142-143 that the CDF might not be known is practice is acceptable is wrong. As demonstrated by the lower bound, even very small differences in between the assumed CDF and actual CDF may lead to $\sqrt{T}$ regret. Since it is not clear how either new algorithm reacts to misspecification of the noise, I do not think that we can claim that not knowing the CDF is acceptable in practice.

I also have a slightly more technical question. I have not seen the requirement that $1-F$ is log-concave before and I wonder if there is a relationship between $F$ being log-concave and $1-F$ being log-concave. If not, is there a straightforward manner in which I can verify that for example a gaussian distribution satisfies the requirement that $1-F$ is log-concave?

Overall, the paper contains some interesting results. Although I doubt the practical relevance of the results I do appreciate the technique to use the log-concavity property to derive surrogate losses. I also appreciate the lower bound an it's implications.

The writing could be improved as there are some unclarities. For example, upon my first reading I read assumption 2 as if the true parameter was known to us. In the sentence starting on line 74 it is unclear to me what "when we explain the results" means. There are also bothersome inconsistencies in the use of big $O$ notation. For example, in line 44 $d$ is included in the big $O$ notation but in line 49 it is not. In line 16 it is not clear to me what an acceptable price is and to me it seems that finding the price that is the most profitable is the goal. The $C$ used in Lemma 8 does not appear to be defined in equation 11.

Minor comments
- I do not understand the "|" symbols in lines 295 and 294.
- line 226: $C$ and $C_{down}$ is defined --> $C$ and $C_{down}$ are defined.
- perhaps the greater than zero is unnecessary in equation (12) due to the definition of semi-positive definite matrices.
- as a personal preference I try to avoid the use of obviously and clearly as whenever I do not understand statements which include either of these words immediately I always feel a little bit stupid. I do not think it would harm the paper if these words are removed so I suggest you remove them.
- line 131: till is perhaps too informal --> until
- line 570: following --> follows
- line 561: curve --> curves
- line 536: In specific --> to be specific.
- line 536: a "." before Define is missing.
- line 537: there is a "." missing at the end of the sentence.


----------------------------------
In the rebuttal the authors have adequately addressed my concerns and I have raised my score.

**Time Spent Reviewing:**

6

---

> ### Author Response · Authors · 2021-08-06
> **Response to Reviewer XCzw**
>
> Thank you for the detailed feedback. In the following part, we will firstly address your main concern regarding the noise distribution and the coefficients it contributes to the regret bounds.
>
> In fact, we have discussed this issue in Section 7 and Appendix C.1. We understand that we did not clearly state how big the constants are in the Gaussian noise case, and we are sorry that we reversely wrote the inequality notations below line 594 by mistake. In fact, the two inequalities should be:
> $$C_{\text{down}}\geq\inf_{\omega\in[-\frac{B}{\sigma}, \frac{B}{\sigma}]} \{\frac{1}{\sigma^2}\lambda(-\omega)^2+\omega\cdot\lambda(-\omega) \}$$
> and
> $$C_{\text{exp}}\leq\sup_{\omega\in[-\frac{B}{\sigma}, \frac{B}{\sigma}]}\{\frac1{\sigma^2}\lambda(-\omega)^2\}.$$
> By combining these two inequalities with the analysis in Lemma 21, we achieve an upper bound of the constant $\frac{C_{exp}}{C_{down}}$, which increases exponentially as $\frac{B}{\sigma}$ goes to infinity. This is because a large σ helps exploration and thus makes the likelihood function smoother. However, as we assumed in Section 3 that B and σ are constants known in advance, the cumulative regret will converge to $O(d\log T)$ as $T$ goes to infinity. In comparison, existing works can be divided into two classes according to their methods dealing with the noise. On the one hand, prior works (Cohen et al., 2016, Krishnamurthy et al., 2020) deal with small-variance noise in a “confidentially cutting” way. The coefficients of their regrets naturally decrease as $\sigma$ goes large. However, this method is only suitable when $\sigma=O(\frac{1}{T})$. On the other hand, Javanmard and Nazerzadeh [2019] deals with constant-variance noise as we do in this work, and they also have an exponential dependence on $\frac{B}{\sigma}$: their regret includes the (quadratic) inverse of a coefficient $l_W$ that defines almost the same as our $C_{down}$. We will add a comparison of coefficients among related literatures, and also explain in detail about those not taken into comparison (e.g., those small-variance noise algorithms).
>
> As for the claim in line 143 that Assumption 1 is still acceptable, we state this mostly according to an information-theoretic perspective: On the one hand, it is *necessary* for a low regret; on the other hand, it has been *sufficiently* used in previous works (e.g. Javanmard and Nazerzadeh [2019]). Although we did not present a method to precisely estimate $\sigma$ in this work, it is a reasonable algorithm to replace $\sigma$ with a plug-in estimator $\hat{\sigma}$ estimated using historical offline data. The reviewer is absolutely right *in a minimax sense*.  As we have shown, not knowing $\sigma$ requires $\sqrt{T}$ regret *in general*, but the lower bound does not rule out the plug-in approach achieving a smaller regret for interesting subclasses of problems in practice. We will add a more detailed discussion of Assumption 1.
>
> We also assume strict log-concavities for both $F$ and $1-F$, which the reviewer asks about. In fact, if the CDF $F$ is strictly log-concave and also its PDF $f$ is symmetric (i.e., $f(-x)=f(x)$), then we have $F(-x)=1-F(x)$. As a result, we have:
> $$\frac{d^2\log{(1-F(x))}}{dx^2}=\frac{-f’(x)(1-F(x))-(f(x))^2}{(1-F(x))^2}=\frac{d^2\log{F(-x)}}{dx^2}.$$
>
> That is to say, symmetric distributions such as Gaussian always satisfy (or not satisfy) the strict log-concavity for $F$ and $1-F$ simultaneously. We hope this explanation clarifies this log-concavity assumption.
>
> ---------------
>
> Finally we thank the reviewer for enumerating typos and/or broken sentences due to editing. We will fix all of them in the final version. Some more detailed clarifications are given below:
>
> (1)  For the big $O(\cdot)$ notation: We formally claim our regret bound as $O(d\log{T})$. When we compare our results to those of other literatures, however, we sometimes drop the dependence on $d$ and form an $O(\log{T})$ regret. This is because of the different assumptions on dimensionality in some literature. For example, in Javanmard and Nazerzadeh, [2019], the author assumes a sparsity factor $s_0$ (which equals $d$ in a non-sparse setting) and includes it in the regret bound as $O(s_0\log d\cdot\log{T})$. Since we do not consider sparsity, the dependence on $s_0$ is replaced by $d$ in our results. Overall, we focus on the dependence on $T$ without making distributional assumptions on $x$.
>
> (2)  In line 16, an “acceptable-and-profitable” price means that the price is neither too high to afford nor too low to make profits. If we take expectations over the probability of acceptance, then what we want is exactly the most profitable price.
>
> (3)  The $C$ used in Lemma 8 is defined in Lemma 5.
>
> (4)  In line 295 and 294, the notation “$a|b$” means “integer $b$ is divisible by integer $a$”. This is a number-theoretic notation, and we promise to use $b\equiv0(\mod a)$ to substitute this notation.
>
> --------------------------------
>
> We hope that our response has appropriately addressed your concerns, and also hope that you kindly consider raising the score. Thank you again for your careful review and valuable input!

---

> > ### Comment · Area_Chair_aYga · 2021-08-20
> > **Log concave curiosity**
> >
> > As a curiosity (unrelated to the paper), I even if the PDF Is not symmetric the log-concavity of the PDF implies log-concavity of both F and 1-F. This is due to Prekopa's inequality.

---

> > > ### Author Response · Authors · 2021-08-24
> > > **Thanks for your comment!**
> > >
> > > We thank the area chair for pointing out this interesting property. This enables our algorithms to be applied in more cases than symmetric distributions.

---

> > > > ### Comment · Reviewer_XCzw · 2021-08-25
> > > > **Post-rebuttal update**
> > > >
> > > > Thank you for the rebuttal, it has addressed my concerns and I will raise my score.

---

> > > > > ### Author Response · Authors · 2021-08-27
> > > > > **Thank you for your support!**
> > > > >
> > > > > We thank you for your carefully reviewing our rebuttal and kindly updating the score!

---

### Official Review · Reviewer_KZQd · 2021-07-19

**Rating:** 6
**Confidence:** 4

**Summary:**

This paper studies the recently active problem of regret minimization in feature-based pricing.


**Limitations And Societal Impact:**

Yes

**Main Review:**

In feature-based pricing, we have a buyer with a high-dimensional value theta that is held privately. Features x_t from the same dimension arrive online, and the value w_t for round t is <theta,x_t> + zero-mean-strictly-log-concave-noise. The noise is drawn i.i.d. from a known bounded strictly log-concave distribution (e.g., Gaussian noise). The buyer is posted a price p_t at round t, and buys when his value is at least the price, and rejects otherwise. The algorithm designer who posts the price gets to see only the features (before posting the price) and the binary decision of sale or no sale (after posting the price). The goal of the seller is to post a stream of prices that minimizes the revenue regret --- compare the obtained revenue with the optimal revenue obtainable had the seller known the hidden vector theta.

The paper studies two settings, one where the features are stochastic (drawn i.i.d. from an unknown distribution) and the other where the features are drawn adversarially. In both the settings, the paper gives algorithms that achieve O(d log T) regret.

I generally like this line of work as it provides a neat playground to develop good theory and techniques. Achieving a regret bound of O(d log T) in this model is a nice/clean result. Knowing the noise is a reasonable assumption and there is anyway regret lower bound of \sqrt{T} when not knowing the noise distribution.

My primary question is regarding the regret bound: what is the dependence on the level of noise in the regret bound of O(d log T)? To take a simple case, with Gaussian noise, when sigma approaches zero, can this paper get a O(d log T) bound? If not, is there a reason to expect that achieving O(d log T) regret is impossible when sigma is zero? A priori, it seems like getting O(d log T) is not impossible. The current writing conceals the dependence on the noise (or specifically sigma for the Gaussian case), and it is hard to figure out from the paper what the dependence on sigma is. Typically one would think that the less noise case is the easier one, and more the noise harder it is. But given the highly discontinuous nature of pricing loss, noise can help "smooth out".

Regardless of the answer, there should be a clear discussion on the dependence on noise in the regret achieved.

Another subtlety to point out is the somewhat unusual notion of regret: usually we talk about E[max], but here we have max E. In other words, rather than compare with the expectation of the ex post optimal policy that posts a price of w_t, the comparison is w.r.t. the price offering the maximum expected reward. While understandably, comparison w.r.t. the former will lead to a \sqrt{T} regret due to the variance induced by the noise, this is an important enough aspect that deserves to be discussed.

**Time Spent Reviewing:**

4

---

> ### Author Response · Authors · 2021-08-06
> **Response to Reviewer KZQd**
>
> We thank you for your careful reading and the insight on the two important issues: the noise level and the regret definition. In the following part, we will carefully explain these two points according to our submission contents.
>
> ----------------
>
> 1, **The dependence of regret on the noise level**: Our upper regret bounds indeed have an "inverse dependence" on the level of noise (see Section 7). To be specific, we have analyzed the case of Gaussian noise in Appendix C.1 and have shown that the coefficient of regret bound (i.e., $\frac{C_{exp}}{C_{down}}$) depends exponentially on $\frac{B}{\sigma}$. This is because a large $\sigma$ helps exploration and thus makes the (conditional expected) log-likelihood function smoother. However, as we assumed in Section 3 that B and σ are constants known in advance, the cumulative regret thus converges to O(d log T) as T goes to infinity. It is worth mentioning that, when the noise variance is extremely small, i.e. $\sigma=O(\frac{1}{T\log{T}})$ (including $\sigma=0$), Cohen et al. [2020] has already settled the problem and achieved an $O(\log{T})$ regret as well.
>
> In a nutshell, the minimax regret for this problem setting is still $O(\log{T})$ if $\sigma$ is close enough to zero, but our algorithms are not designed for that case: likelihood functions do not exist when $\sigma=0$. As we pointed out at the end of Section 7, the existence of an algorithm that guarantees $O(\log{T})$ regret for any $\sigma$ is still an open problem. To make the algorithm more practical at this stage, we might apply the "ShallowPricing" algorithm (see Cohen et al. [2020] Thm. 3) as a pre-processing step to localize the learner such that $\frac{B}{\sigma}$ is small.
>
> Please also see our response to Reviewer VSt3 for the intuition behind why noise helps.
>
> -----------
>
> 2, **The difference between *ex ante* and *ex post* regrets**: We thank the reviewer for asking this very interesting question. In this work, we considered the *ex ante* regret $Reg_{ea}=\sum_{t=1}^T\max_{\theta}\mathbb{E}[v_t^{\theta}\cdot1(v_t^{\theta}\leq{w_t})]-\mathbb{E}[v_t\cdot1(v_t\leq{w_t})]$, where $v_t^{\theta}=J(x_t^{\top}\theta)$ is the greedy price with parameter $\theta$ and $w_t=x_t^\{\top}\theta^*+N_t$ is the realized random valuation. In comparison, an *ex post* definition of the cumulative regret, i.e., $Reg_{ep} = \max_{\theta}\sum_{t=1}^Tv_t^{\theta}\cdot1(v_t^{\theta}\leq{w_t})-v_t\cdot1(v_t\leq{w_t})$ also makes sense. Note that we can decompose $\mathbb{E}[Reg_{ep}]=Reg_{ea}+\mathbb{E}[\max_{\theta}\sum_{t=1}^{T}v_t^{\theta}\cdot1(v_t^{\theta}\leq{w_t})-\sum_{t=1}^{T}v_t^{\theta^*}\cdot1(v_t^{\theta^*}\leq{w_t})]$, where the second term totally depends on the problem setting and is not relevant to the algorithm (although it could be as large as $\Omega(\sqrt{dT})$ as the reviewer points out). Therefore, we can use $Reg_{ea}$ as the regret metric without losing generality. For more detailed discussion on the *ex ante* and *ex post* regrets, please see our Appendix C.4.
>
> On a higher level, max E Regret had been considered in various stochastic bandit settings (k-arm / linear / contextual)  and maybe it is not that esoteric?
>
> ------------
> We hope that our response has appropriately addressed your concerns and you could kindly consider raising the score.  Thank you again for your input!

---

### Official Review · Reviewer_VSt3 · 2021-07-21

**Rating:** 7
**Confidence:** 4

**Summary:**

(Disclaimer: I have previously reviewed this paper for ICML 2021).

This paper develops algorithms for the problem of contextual dynamic pricing. This is an online learning problem that takes place over T rounds. There is a customer with a hidden value vector theta in R^d. Every round, the learner is given as context a feature vector x in R^d (both x and v are suitably bounded), representing the features of the current item. The customer is willing to pay up to v = <x, theta> + a Gaussian noise term for the item. The learner must set a price p for the item this round; if p > v the customer does not buy the item (and the learner receives 0 utility), otherwise the customer buys the item at price p and the learner receives p utility. The goal of the learner is to maximize their total learner (minimizing their regret compared to the optimal dynamic policy).

This paper presents two algorithms for two different settings. Their first algorithm (EMLP) applies when the feature vectors x_t are iid drawn from an unknown distribution. Their second algorithm (ONSP) applies when the feature vectors x_t are chosen adversarially. In both cases, they obtain O(d*log T) regret. Both algorithms require that the noise is Gaussian with a known variance sigma^2 (The authors show an O(sqrt(T)) lower bound if sigma is unknown). (In fact, this known variance sigma influences the constant in the O(d*log T) regret; see comments below).

Roughly, both algorithms work by maintaining an estimate of theta, and submitting the expected-utility-maximizing price p (assuming the theta estimate is accurate) per round. The two algorithms differ in how they maintain their estimate of theta. EMLP divides the rounds into log T geometrically-sized epochs, and uses a MLE to recompute the value of theta at the end of each epoch. ONSP instead runs an online Newton step method each round to obtain an estimate of theta.

The authors additionally present some empirical simulations of their algorithms (on synthetic data), showing that they outperform EXP4.

**Limitations And Societal Impact:**

No concerns here.

**Main Review:**

I am generally positive about this paper: I think contextual pricing is a pretty interesting and important question, and I think this paper makes concrete and novel progress in developing algorithms in a noisy setting.

One concern from the previous submission of this paper was that it was missing important citations in the literature (namely a line of a recent work on contextual pricing and contextual search). The paper now contains a good discussion of how this paper significantly differs from the models considered in these papers (and as far as I can tell, I agree).

One other concern was that the regret of the authors’ algorithms depended in a significant (and non-intuitive way) on sigma -- in particular, as sigma gets small (less noise), the regret of their algorithms blow up. This was not made very clear in the earlier submission, but in the current draft there is now some discussion of this in section 7. This phenomenon still somewhat bothers me; it feels in some sense that the algorithms the authors present are mostly taking advantage of how the noise smooths out the reward function rather than dealing with the core difficulties of dynamic pricing, but I admit that this is still an interesting regime.

The paper was well-written and easy to read.


**Time Spent Reviewing:**

1.5

---

> ### Author Response · Authors · 2021-08-06
> **Response to Reviewer VSt3**
>
> Thank you for the encouraging feedback! We hope that our new submission here has so far addressed your concerns in the previous review. The paper is stronger and clearer thanks to your input! We appreciate it.
>
> Your comment on the “non-intuitive” dependence on $\sigma$ is valid. Let us explain the intuition behind it.  First let us zoom out a bit and inspect the problem setting in a broader context. It is an online learning problem with a special bandit feedback. Without full information feedback, exploration is needed and it is usually impossible to obtain sub $\sqrt{T}$ regret. Note that $\Omega(\sqrt{T})$ regret is required even if we make the nicest assumption (e.g. strongly smooth and strongly convex). Our algorithm is not exploring (neither using optimism nor random sampling) at all yet it gets away with an $O(\log{T})$ regret. The reason is that the noise in the problem itself  helps the learner with exploration implicitly. Thus, your observation is accurate and our algorithm is indeed taking advantage of the noise to do better, thus the seemingly non-intuitive dependence. On a very high level, the fact that we can sometimes take advantage of the stochasticity in the data for exploration is an interesting observation in its own right and the same phenomena could be seen elsewhere.
>
> The very low-noise setting that was considered in existing contextual search /pricing literature is an easier problem as bisection can be used to obtain logarithmic regret naturally. Similar bisection algorithms can be applied when $B>>\sigma$ to localize the learner (because the feedback selects the correct halfspace w.h.p.) to a much smaller region where B is comparable to $\sigma$; then a natural idea is to switch to our EMLP or ONSP algorithms after the localization step.  We tried to formalize this idea but ran into some technical difficulties.

---

### Author Response · Authors · 2021-08-06
**Response to all reviewers**

We thank all reviewers for their efforts on reviewing our work and their valuable suggestions/comments! In the following part, let us carefully address each of these concerns respectively.

---

### Decision · Program_Chairs · 2021-09-27

**Decision:**

Accept (Spotlight)

**Comment:**

All the reviewers agree on the importance of the contribution. It is also my view that obtaining logarithmic regret is an important open problem in contextual pricing without any assumption of small noise. I am aware of many papers that tried to establish similar results but failed.

The only concern shared by the reviewers is that the regret bound is inversely proportional to noise, but this was  adequately addressed this in the rebuttal by saying that larger noise leads to more exploration. The authors are encouraged to make this more explicit in the revision.

The review team is confident that this should be one of the top papers in NeurIPS and enthusiastically recommends acceptance.